# HAICO-CN: Human-AI Collaboration By Cluster-wise Noisy-Label Augmentation

## Abstract

The intricate dynamics of human-AI collaboration presents an ongoing challenge. While recent research incorporates human behaviors into machine learning model design, most utilise single global confusion matrix or human behavior model, disregarding potential personalization to user. To address this gap, we propose HAICO-CN, a human-AI collaborative method that enhances human-AI joint decision-making by training personalized models using a novel cluster-wise noisy-label augmentation technique. During training, HAICO-CN first identifies and clusters noise label patterns within the multi-rater data sets, followed by a cluster-wise noisy-label augmentation method that generates enough data to train a collaborative human-AI model for each cluster. During inference, the user follows an onboarding process, allowing HAICO-CN to select a cluster-wise human-AI model based on the user's noisy label patterns, thereby enhancing human-AI joint decision-making performance. HAICO-CN is simple to implement, model-agnostic, and effective. We propose new evaluation criteria for assessing human-AI collaborative methods and empirically evaluate HAICO-CN across diverse datasets, including CIFAR-10N, CIFAR-10H, Fashion-MNIST-H, and Chaoyang histopathology, demonstrating HAICO-CN's superior performance compared to state-of-the-art human-AI collaboration approaches.

## 1 Introduction

Determining the optimal human-AI collaboration mechanism has been challenging due to its multifaceted nature (Dafoe et al., 2021). On one side, humans decisions tend to be driven by creativity and innovation and often incorporate contextual understanding. Nevertheless, these decisions are susceptible to errors and biases, even among professionals in critical fields like healthcare (Scott & Crock, 2020). On the other side, machine learning (ML) models, while increasingly matching or surpassing human accuracy in various specialised tasks, often lack contextual knowledge, commonsense reasoning, and the capability of applying complex problem-solving heuristics (Holstein & Aleven, 2021; Lake et al., 2016; Miller, 2019). The research on human-AI collaboration seeks to harness the strengths of both humans and AI to achieve optimal effectiveness and efficiency for human-AI joint decision-making.

The research community has proposed a variety of innovative human-AI collaboration mechanisms and frameworks, such as learning to defer (Raghu et al., 2019; Madras et al., 2018), learning to complement (Wilder et al., 2021), human-in-the-loop (Wu et al., 2022), and algorithm-in-the-loop (Green & Chen, 2019). Notably, recent works have taken human behaviors into account when designing machine learning models. For example, Vodrahalli et al. (2022) improves the final human decision by deliberately displaying uncalibrated confidence values derived from a human behaviour model, which prompts human to rectify otherwise incorrect decisions. Kerrigan et al. (2021) combines the human prediction uncertainty and AI model probability to improve the accuracy of machine learning systems. However, these approaches utilise only one single global confusion matrix or human behavior model. It remains uncertain whether these frameworks can maintain their performance when used by a diverse range of users, considering that the human-AI collaboration performance could vary significantly based on individuals' expectations, knowledge, or experience levels (Kocielnik et al., 2019; Wang et al., 2021).

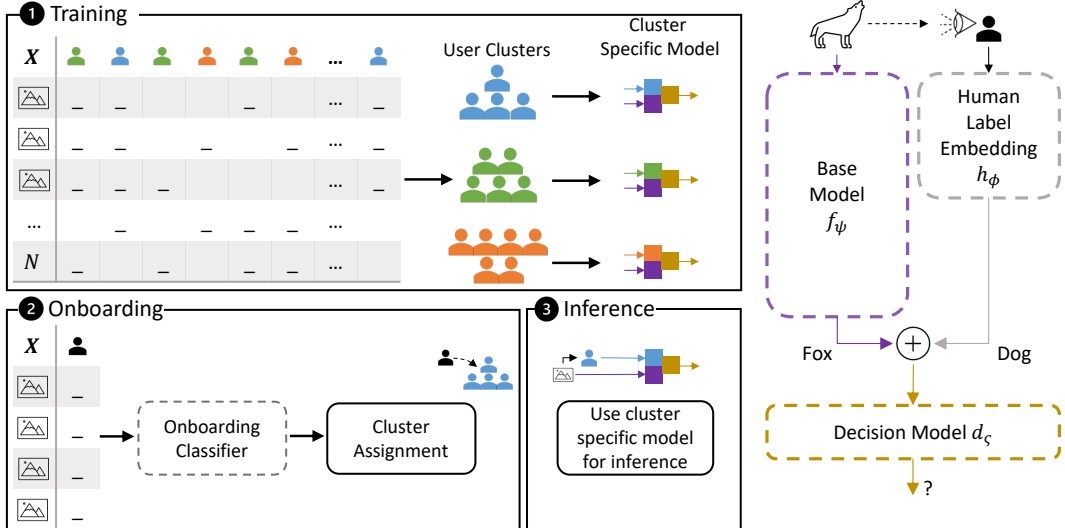

Figure 1: Training and inference of HAICO-CN.

Figure 2: Model architecture of HAICO-CN.

To address this knowledge gap, we propose a novel human-AI collaborative method, HAICO-CN, that enhances joint human-AI decision-making by training personalized models for different user groups using a novel cluster-wise noisy-label augmentation method (Figure 1). More specifically, given a multi-rater training set, during the training phase, HAICO-CN first clusters raters into multiple groups, each having distinct noisy label patterns. Subsequently, we employ a novel noisy-label augmentation approach to generate sufficient number of labels with specific noise patterns for training cluster-wise human-AI collaboration models. During the inference phase, the user first undergoes an onboarding process where the user provides labels to a set of images. After that, a cluster-wise human-AI collaboration model is selected based on the user's unique noise pattern, resulting in state-of-the-art accuracy across various human-AI joint decision making tasks. The proposed framework is model agnostic and can be trained with noisy labels from multiple raters without the need for ground truth labels. Hence, the main contributions of this paper are:

- A novel human-AI collaborative (HAICO-CN) method that leverages predictions from both human experts and AI models, personalizing to individual user patterns.
- A noisy-label augmentation technique to generate a sufficient number of noisy labels for training the cluster-wise human-AI collaboration models.
- An innovative experiment setup featuring both synthetic and public multi-rater datasets (CIFAR-10N, CIFAR-10H, Fashion-MNIST-H, and Chaoyang histopathology) with new evaluation criteria for assessing human-AI collaborative frameworks.

Empirical studies across our datasets consistently demonstrate HAICO-CN's superior performance compared to the state-of-the-art human-AI collaboration methods in various classification tasks.

## 2 RELATED WORK

The traditional notion that increased automation leads to decreased human control (Parasuraman et al., 2000; Committee, 2014) has been up for re-evaluation in recent years. This simplified perspective, while beneficial in some contexts, often fails to capture the complex interplay between humans and autonomous systems. For example, uncertainties in autonomous systems often demand increased human supervision (Strauch, 2018). As current AI models match or surpass human accuracy in various specific tasks, the research community has proposed novel human-AI collaboration methods, which can be broadly classified into three categories based on the objectives of the models.

**Learning-to-assist** approaches aim to support human decision-making with AI model predictions. These approaches are commonly seem in critical domains, such as law(Liu et al., 2021) and

medicine(Levy et al., 2021), where humans make the final decision. Extensive efforts have been devoted to increase the explainability and transparency of such models (Tjoa & Guan, 2021).

**Learning-to-defer** approaches allow AI models to autonomously manage confident cases and defer decisions to humans when confidence is low (Madras et al., 2018; Mozannar et al., 2023). These approaches focus on the optimization of a utility function that takes into account the accuracy of the AI model, the preference of the human decision maker, and the cost of deferring decisions. For example, Raghu et al. (2019) proposed an ensemble of AI models to predict the risk of death for each patient, and then defers decisions to a human expert for patients with the highest risk of death.

**Learning-to-complement** models are optimized to leverage the strengths from both human and AI model to improve decision-making. For example, Steyvers et al. (2022) proposed a Bayesian framework for modeling human-AI complementarity. Kerrigan et al. (2021) used the calibrated confusion matrix to combine human and model predictions in a way that minimizes the expected loss. Wilder et al. (2021) consider the uncertainty from AI models and humans to jointly train a model that allocate tasks to the AI model or the human to maximize the overall accuracy. HAICO-CN also falls into this category and aims to utilise complementary strengths of both human and AI to arrive at the final decision.

### 2.1 HUMAN BEHAVIOUR

Recent papers have taken human behavior into account when designing the human-AI collaboration models. Green & Chen (2019) found that humans were unable to identify AI model's mistakes and sometimes trust AI decisions more than they trusted their own judgement. The paper argues that humans are often unable to provide the required oversight to ensure that AI models are used responsibly and ethically. Following from this conclusion, Vodrahalli et al. (2022) aimed to make the AI model uncalibrated with the goal of improving the overall accuracy of the final human decision-making. Kerrigan et al. (2021) incorporated global confusion matrices representing human uncertainty to improve the joint human-AI performance in classification tasks. In contrast to previous papers, HAICO-CN aims to leverage the differences among users' noisy patterns in the design of personalised human-AI collaborative models.

### 2.2 EVALUATING HUMAN-AI COLLABORATION

Human-AI complementarity is defined in Dellermann et al. (2021) as leveraging the unique capabilities of both humans and AI to achieve better results than each one could have achieved separately. However, assessing the interaction between humans and AI is intricate, and numerous benchmarks have been suggested in existing literature. In the context of learning-to-assist or learning-to-complement, traditional evaluation criteria such as accuracy, precision, recall are commonly used. For learning-to-defer, measures such as coverage are employed to evaluate the model's performance on the data that is not deferred (Raghu et al., 2019). When dealing with noisy labels, additional measurements such as label precision, label recall, and correction error are also used (Song et al., 2022a). To evaluate the proposed AI model, we introduce an innovative experiment setup with both synthetic and real-world multi-rater datasets and new evaluation criteria (Section 4).

## 3 METHODOLOGY

This section provides details about the consensus label estimation, model training and inference of HAICO-CN. The multi-rater training set for the multi-class image classification task is defined by $\tilde{\mathcal{D}} = \{(\mathbf{x}_i, \{\tilde{\mathbf{y}}_{i,j}\}_{j \in \mathcal{A}})\}_{i=1}^N$, where $\mathbf{x}_i \in \mathcal{X}$ is a data sample, $\tilde{\mathbf{y}}_{i,j} \in \mathcal{Y} \subset \{0,1\}^C$ is a one-hot vector for the $C$-class annotation representing the noisy-label provided by annotator $j \in \mathcal{A}$. Each data sample has a latent clean label denoted by $\mathbf{y}_i \in \mathcal{Y}$. Assuming a class-dependent label noise (or asymmetric noise) (Song et al., 2022b), we explain in the sections below the training, onboarding, and inference stages of HAICO-CN depicted in figure 1.

### 3.1 CONSENSUS LABEL ESTIMATION

The first training stage is the estimation of the consensus label $\bar{\mathbf{y}}_i$ (that approximates the latent clean label $\mathbf{y}_i$) with Crowdlab (Goh et al., 2023) using the noisy labels $\{\tilde{\mathbf{y}}_{i,j}\}_{j \in \mathcal{A}}$ for each training

sample. Crowdlab works in two steps. In the first step, it estimates a consensus by majority vote $\bar{\mathbf{y}}_i'$ per training sample. In the second step, it trains a classifier using the initial consensus and obtains predicted class probabilities for each training example. After that, Crowdlab uses these predicted probabilities along with the original annotations from raters to estimate a better consensus, creating an ensemble as,

$$\bar{\mathbf{y}}_i = \mathbf{w}_\gamma \times f_\gamma(\mathbf{x}_i) + \mathbf{w}_1 \times \tilde{\mathbf{y}}_{i,1} + ... + \mathbf{w}_{|\mathcal{A}|} \times \tilde{\mathbf{y}}_{i,|\mathcal{A}|}, \tag{1}$$

where $f_\gamma : \mathcal{X} \to \Delta^{C-1}$ is a classifier trained with the majority vote label $\bar{\mathbf{y}}_i'$ to output a categorical distribution for $C$ classes, and the weights $\mathbf{w}_\gamma, \mathbf{w}_1, ..., \mathbf{w}_{|\mathcal{A}|}$ are assigned according to an estimate of how trustworthy the model is compared to each individual annotator. The outcome of Crowdlab is a consensus labelled training set denoted by $\tilde{\mathcal{D}} = \{(\mathbf{x}_i, \bar{\mathbf{y}}_i)\}_{i=1}^N$. Note that the consensus label is necessary only when the clean label $\mathbf{y}_i$ is latent. If such clean label is observed, then Crowdlab is no longer needed, and the next stage of our HAICO-CN can be trained with $\mathcal{D} = \{(\mathbf{x}_i, \mathbf{y}_i)\}_{i=1}^N$.

## 3.2 Model Training

As shown next, the training of HAICO-CN has two steps: 1) user clustering using their noisy-label patterns, and 2) training of models for each cluster from step 1 using noisy-label augmentation.

### 3.2.1 User Clustering

As annotators label a subset of training images, the provided labels belong to different images. Yet, to cluster all annotators together, each label set is arranged in a uniform format as presented as in equation 2 to enable the clustering algorithm to determine noisy label patterns present. We take each annotator $j \in \mathcal{A}$ and each class $c \in \{1, ..., C\}$ to build the set of sample labels that have consensus label $c$, with $\mathcal{S}_j^{(c)} = \{\tilde{\mathbf{y}}_{i,j} | (\mathbf{x}_i, \tilde{\mathbf{y}}_{i,j}) \in \tilde{\mathcal{D}}, c = \arg\max_{\tilde{c} \in \{1,...,C\}} \bar{\mathbf{y}}_i(\tilde{c})\}$. We can then build the $L \times C$ vector

$$\mathbf{s}_j = [l_1^{(1)}, ..., l_L^{(1)}, ..., l_1^{(C)}, ..., l_L^{(C)}] \tag{2}$$

for annotator $j \in \mathcal{A}$ that labels $L$ images for each class, where $l_l^{(c)} = \arg\max_{\tilde{c} \in \{1,...,C\}} \tilde{\mathbf{y}}_{i,j}(\tilde{c})$ with $\tilde{\mathbf{y}}_{i,j} \in \mathcal{S}_j^{(c)}$ representing one of the noisy labels from $\mathcal{S}_j^{(c)}$. $\mathbf{s}_j$ may be different, but class order is preserved for all annotators. This process is repeated for all annotators to form the set $\mathcal{L} = \{\mathbf{s}_j\}_{j \in \mathcal{A}}$.

The clustering process employs Fuzzy K-Means using the set of vectors $\mathcal{L}$ to automatically identify groups of annotators who have similar noisy-label patterns (Dehariya et al., 2010). Unlike K-Means, which tries to assign at least one data point per cluster, Fuzzy K-Means allows clusters to be empty, which in turn helps mitigate potential issues arising from not precisely fine-tuning the number of clusters $K$. Fuzzy K-Means runs until stable clusters are found. After clustering, each user is associated with a particular cluster $k \in \{1, ..., K\}$ that contains users with similar noisy-label patterns.

### 3.2.2 Noisy-label Augmentation

After clustering, the original training set $\tilde{\mathcal{D}}$ is divided into $K$ subsets $\tilde{\mathcal{D}}_k \subset \tilde{\mathcal{D}}$, each containing the users allocated to cluster $k$. Unfortunately, these subsets may not contain all data samples from the original training set given that users may not have annotated all samples in $\mathcal{D}$. Given such a small amount of data to train the models for each cluster, we propose a data and label augmentation process for each cluster. Such label augmentation is obtained by sampling from the estimated cluster-specific label transition from the consensus label to the noisy label. An advantage of our proposed noisy-label augmentation is the reinforcement of the label biases found in each cluster which facilitates the training of a classifier to be robust to such biases.

Let us assume that cluster $k$ has a subset of the annotators denoted by $\mathcal{A}_k \subset \mathcal{A}$. This will allow us to estimate the cluster $k$'s label transition matrix $\mathbf{T}_k \in [0, 1]^{C \times C}$ as follows

$$\mathbf{T}_k(c, :) = \frac{1}{|\mathcal{A}_k|} \sum_{\tilde{\mathbf{y}}_i \in \left\{\mathcal{S}_j^{(c)}\right\}_{j \in \mathcal{A}_k}} \tilde{\mathbf{y}}_i, \tag{3}$$

where $\left\{ \mathcal{S}_j^{(c)} \right\}_{j \in \mathcal{A}_k}$ denotes the set of labels, from samples with consensus label $c$, for all users in $\mathcal{A}_k$. Note that each element of the transition matrix for cluster $k$ from equation 3 denotes the probability that a user in cluster $k$ flips from the consensus label $\bar{Y} = c$ to the noisy label $\tilde{Y} = n$, as in $\mathbf{T}_k(c,n) = p(\tilde{Y} = n | \bar{Y} = c, R = k)$, where $R$ is the random variable for the user cluster. For each data point $\mathbf{x}_i$ in $\tilde{\mathcal{D}}_k$, we take its consensus label $c = \arg\max_{\tilde{c} \in \{1,...,C\}} \bar{\mathbf{y}}_i(\tilde{c})$ computed from equation 1 and the cluster $k$'s transition matrix $\mathbf{T}_k$ from equation 3 to generate $G$ labels by sampling $\{\hat{\mathbf{y}}_g\}_{g=1}^G \sim p(\tilde{Y} | \bar{Y} = c, R = k)$, which represents the categorical distribution in row $c$ of the transition matrix $\mathbf{T}_k$. The new noisy-label augmented training set for each cluster $k$ is denoted by $\hat{\mathcal{D}}_k = \{(\mathbf{x}, \{\hat{\mathbf{y}}_g\}_{g=1}^G) | (\mathbf{x}, \{\tilde{\mathbf{y}}_j\}_{j=1}^{A_k}) \in \tilde{\mathcal{D}}_k, \{\hat{\mathbf{y}}_g\}_{g=1}^G \sim p(\tilde{Y} | \bar{Y} = c, R = k)\}$.

### 3.2.3 MODEL ARCHITECTURE AND TRAINING

The proposed architecture, shown in figure 2, has three components, namely: 1) a base model that transforms input data into a logit with $f_{\psi_k} : \mathcal{X} \to \mathbb{R}^C$; 2) a human label embedding that takes the one-hot user provided noisy label and transforms it into a logit with $h_{\phi_k} : \mathcal{Y} \to \mathbb{R}^C$; and 3) a decision model that takes the model's and human's logits to output a categorical distribution with $d_{\zeta_k} : \mathbb{R}^C \times \mathbb{R}^C \to \Delta^{C-1}$. The whole model $m_{\theta_k} : \mathcal{X} \times \mathcal{Y} \to \Delta^{C-1}$ is defined as:

$$m_{\theta_k}(\mathbf{x}, \hat{\mathbf{y}}) = d_{\zeta_k}(f_{\psi_k}(\mathbf{x}) \oplus h_{\phi_k}(\hat{\mathbf{y}})), \tag{4}$$

where $\theta_k = \{\psi_k, \phi_k, \zeta_k\}$, and $\oplus$ represents the concatenation operator. In equation 4, the base model $f_{\psi_k}(.)$ can be any model trained on $\bar{\mathcal{D}}$. The human label embedding model $h_{\phi_k}(.)$ aims to discover the label biases of cluster $k$, and $d_{\zeta_k}(.)$ aims to model the joint label noise distribution between the base model and human label embedding to make $m_{\theta_k}(\mathbf{x}, \hat{\mathbf{y}})$ robust to label noise. $h_{\phi_k}(.)$ and $d_{\zeta_k}(.)$ can have any design, but in our experiments, we configured them to be a multi-layer perceptron with two linear layers, and three linear layers (with ReLU activations), respectively.

The model in equation 4 is trained using the following optimisation:

$$\{\theta_k^*\}_{k=1}^K = \arg\min_{\{\theta_k\}_{k=1}^K} \frac{1}{K \times |\hat{\mathcal{D}}_k| \times G} \times \sum_{k=1}^K \sum_{\left(\mathbf{x}_i, \{\hat{\mathbf{y}}_{i,g}\}_{g=1}^G\right) \in \hat{\mathcal{D}}_k} \ell\left(\bar{\mathbf{y}}_i, m_{\theta_k}(\mathbf{x}_i, \hat{\mathbf{y}}_{i,g})\right) +$$
$$\lambda \times \ell\left(\hat{\mathbf{y}}_{i,g}, (\mathbf{T}_k)^\top \times m_{\theta_k}(\mathbf{x}_i, \hat{\mathbf{y}}_{i,g})\right), \tag{5}$$

where $\bar{\mathbf{y}}_i$ is the consensus label from equation 1, $\ell(.)$ is the cross-entropy loss, $\lambda \in [0, \infty]$ is a hyperparameter, and the second loss term is motivated by the forward correction procedure proposed by Patrini et al. (2017), which transforms the clean label prediction from $m_{\theta_k}(.)$ into the noisy label from the noisy-label augmented training set $\hat{\mathcal{D}}_k$ using the transition matrix $\mathbf{T}_k$.

### 3.3 TESTING USING ON-BOARDING PERSONALIZATION

The personalization to testing users involves a two-step on-boarding procedure. The first step consists of classifying the testing user into one of the $K$ clusters, so the optimal classifier $m_{\theta_k}(.)$ is used. The second step sets an on-boarding criterion based on a comparison between the accuracy of the testing user and the base model $f_{\psi_k}(.)$.

The classifier used in the first step is trained with samples that consist of randomly collected labels of $M$ training samples for each of the $C$ classes (estimated from the consensus labels), from users belonging to each of the $K$ clusters. This forms multiple vectors of size $M \times C$, which have the structure defined in equation 2, where each of those vectors is labelled with the user's cluster. We then train a one-versus-all (OVA) support vector machine (SVM) $K$-class classifier. For classifying a testing user into one of the $K$ classes, we need to build an $M \times C$ vector to be classified by the OVA SVM classifier. To build such vector, we use a validation set denoted by $\mathcal{V} = \{(\mathbf{x}_i, \mathbf{y}_i)\}_{i=1}^{M \times C}$, whose images do not overlap with images belonging to the training and testing sets. We request the testing user to label each one of the images in the validation set, so we can build the testing sample to classify the testing user into cluster $k \in \{1, ..., K\}$. For the second step, we compare the model accuracy and testing user accuracy in the validation set $\mathcal{V}$, and only allow the use of the model $m_{\theta_k}(.)$ if the base model $f_{\psi_k}(.)$ accuracy is higher than the testing user accuracy (Steyvers et al., 2022). Testing with $m_{\theta_k}(.)$ is then performed in the testing set $\mathcal{T} = \{(\mathbf{x}_i, \mathbf{y}_i)\}$, whose images do not overlap with images belonging to the training and validation sets.

## 4 EXPERIMENTS

We present a comprehensive analysis of HAICO-CN through a CIFAR-10 (Krizhevsky, 2009) simulation and experiments on public datasets including CIFAR-10N (Wei et al., 2022), CIFAR-10H (Peterson et al., 2019), Fashion-MNIST-H (Ishida et al., 2023), and Chaoyang (Zhu et al., 2022).

### 4.1 DATASETS

**CIFAR-10** comprises 50,000 training, 200 validation, and 9,800 testing class-balanced color images, each sized $32 \times 32$ and belonging to one of 10 classes. **CIFAR-10N** extends the training set of CIFAR-10 by crowd-sourcing its labelling to a pool of 747 annotators, ensuring that each image has exactly three labels, each produced by a distinct annotator. Although annotators provide a minimum of 10 and a maximum of 3070 labels, the majority produces around 200 labels. Similarly, **CIFAR-10H** extends the CIFAR-10 testing set by crowd-sourcing it to be labelled by 2571 annotators, each contributing 210 labels. The resulting label set contains a range of 50 to 63 labels per image from distinct annotators, averaging at around 51 labels per image. Fashion-MNIST (Xiao et al., 2017) is a dataset that originally comprises 60,000 training samples, and 10,000 testing samples with class-balanced images (belonging to one of 10 classes) of size $28 \times 28$ pixels. However, we use **Fashion-MNIST-H**'s multiple annotations on the testing set of Fashion-MNIST for training the model, which means that we use the 10,000 original testing images to train the model, and we divide the original training set into 200 validation images and 59,800 testing images. **Fashion-MNIST-H**'s 10,000 images are annotated by 885 annotators, where the number of labels provided per annotator varies from 9 to 9999, with an average of 753 per annotator. Each image has 66 labels on average with a minimum of 53 and a maximum of 79. Lastly, **Chaoyang** is a pathological dataset featuring four classes of images, having a training set of 4021 images, a validation set with 80 images, and a testing set of 2059 images. Notably, each image in the training set is labeled by three experts, resulting in three labels per image, while the testing set presents a single consensus label.

### 4.2 EXPERIMENTAL SETUP

The simulation experiments on CIFAR-10 consists of a pairwise flipping experiment, where 8 out of 10 classes have 100% of clean labels, but in two classes, 80% of samples have labels flipped to the incorrect class. We simulate three clusters of users, one that flips 80% of the samples between classes airplane↔bird, another cluster that flips horse↔deer, and the other cluster that flips truck↔automobile. For each cluster, we simulate five training and five testing users, producing a total of $5 \times 3 = 15$ unique users for training and another 15 users for testing. The training images, together with the 15 labels/image by the training users, will form $\tilde{\mathcal{D}}$. The model for each cluster $k$, $m_{\theta_k}(.)$ in equation 4, uses a ResNet-18 He et al. (2016) as $f_{\psi_k}(.)$, where the number of clusters is a hyper-parameter since this information is not assumed to be available during the training.

When training with CIFAR-10N, we present two experiments. For the first experiment, the labels from 747 annotators form $\tilde{\mathcal{D}}$. Out of them, 155 were identified for having annotated at least 20 images per class, and they were split in half, taking 79 as training users and 80 as testing users. The training users' labels are used to build the $K$ clusters and to train the OVA SVM classifier. During testing, a testing user's noisy-label transition matrix is estimated using the annotator's labels and consensus labels. This matrix is used to simulate noisy annotations from that testing user. Therefore, 80 noisy test sets are produced, with each representing the biases that each user possesses. The model for each cluster $k$, denoted by $m_{\theta_k}(.)$, uses ViT-Base-16 (Dosovitskiy et al., 2020) as the backbone for $f_{\psi_k}(.)$. For the second CIFAR-10N experiment, we use CIFAR-10H as the testing set, where the labels from testing users were used without any modification for on-boarding. The same labels were used to estimate a noise transition matrix and simulate their own test set. For all 2571 users, their own test test was simulated with own biases. As CIFAR-10N and CIFAR-10H have human labels for CIFAR-10 training and testings sets, respectively, and as they are separately collected datasets, this experiment can be considered as close real world experiment to validate our approach. The models trained for CIFAR-10N were used for this experiment.

For the Fashion-MNIST-H experiment, the labels from all 885 annotators are taken to form the $\tilde{\mathcal{D}}$. Then, 366 out of 885 users are chosen since they have annotated at least 20 images per class and are split in half to have 183 users for training and 183 for testing. The training users' labels are

used to build the $K$ clusters and to train the OVA SVM classifier. During testing, the testing user's noisy-label transition matrix is estimated using the annotator's labels and consensus labels. This matrix is used to simulate noisy annotations from that testing user. Therefore, 183 noisy testing sets are produced, with each representing the biases that each user possesses. The model for each cluster $k$, represented by $m_{\theta_k}(.)$ uses DenseNet-121 (Huang et al., 2017) for $f_{\psi_k}(.)$.

Chaoyang has three annotators per image, which form the $\tilde{\mathcal{D}}$. Training users are clustered and for each cluster $k$, a model $m_{\theta_k}(.)$ is trained with a ViT-Large-16 as the backbone for $f_{\psi_k}(.)$. We simulated noisy testing users using the label noise transition matrices calculated for each annotator, which allows us to produce three distinct testing sets that are subsequently used for evaluation.

It is important to acknowledge that while our method aims to retain noise patterns of annotators, the test sets for Fashion-MNIST-H and Chaoyang remains as simulations that may differ from actual annotator inputs. Our experimental setup with CIFAR-10N and CIFAR-10H datasets more accurately reflects real-world conditions, as these datasets incorporate crowd-sourced labels in both training and testing phases.

For the experiments explained above, we use different backbone models to demonstrate the robustness of our model to such different backbones. To reinforce this point, we show an ablation study that switches the original ViT-Base-16 (Dosovitskiy et al., 2020) to DenseNet-121 (Huang et al., 2017) and Resnet-50 He et al. (2016) on CIFAR-10N in appendix I.

In our CIFAR experiments, we adopted the data augmentation policy introduced by Cubuk et al. (2019). Also, for Fashion-MNIST, alongside random horizontal and vertical flips, we integrated auto augmentations as proposed by Cubuk et al. (2020). For the Chaoyang dataset, data augmentation was limited to random resized crops of dimensions $224 \times 224$. We rely on pre-trained models for $f_{\psi_k}$ because of their robustness to noisy labels (Jiang et al., 2020) (e.g., ViT models were pre-trained on ImageNet-21K, while ResNet-18 and DenseNet-121 models were pre-trained on ImageNet-1K). Adam optimizer was employed for training $f_{\psi_k}(.)$ with consensus $\hat{\mathcal{D}}$, where NAdam was used for training $m_{\theta_k}(.)$ on $\hat{\mathcal{D}}$, each utilizing their respective default learning rates. All implementations were carried out using PyTorch and executed on an NVIDIA GeForce RTX 4090 GPU.

### 4.3 EVALUATION CRITERIA

Our new evaluation criteria are introduced to assess how the user performance is affected by the label alterations made by the model. We first define the positive and negative alteration measures:

$$
\text{Positive Alteration} : A_+ = \frac{1}{|\mathcal{T}| \times |\mathcal{A}|} \sum_{i=1,j=1}^{|\mathcal{T}|,|\mathcal{A}|} \frac{\ddot{\mathbf{y}}_{i,j} = \bar{\mathbf{y}}_i}{\tilde{\mathbf{y}}_{i,j} \neq \bar{\mathbf{y}}_i}
\qquad
\text{Positive Alteration Rate} : R_{A_+} = \frac{A_+}{A_+ + A_-}
$$

$$
\text{Negative Alteration} : A_- = \frac{1}{|\mathcal{T}| \times |\mathcal{A}|} \sum_{i=1,j=1}^{|\mathcal{T}|,|\mathcal{A}|} \frac{\ddot{\mathbf{y}}_{i,j} \neq \bar{\mathbf{y}}_i}{\mathbf{y}_{i,j} = \bar{\mathbf{y}}_i}
\qquad
\text{Negative Alteration Rate} : R_{A_-} = \frac{A_-}{A_+ + A_-}
$$

$$(6) \qquad (7)$$

where $\ddot{\mathbf{y}}_j = \mathsf{OneHot}(m_{\theta_k}(\mathbf{x}, \tilde{\mathbf{y}}_j))$, with the function $\mathsf{OneHot} : \Delta^{C-1} \to \mathcal{Y}$ returning a one-hot label representing the class with the largest prediction from the model $m_{\theta_k}(.)$. In equation 6, $A_+$ quantifies the effectiveness of the model to correct users' labels, where the user provided incorrect labels. In contrast, $A_-$, in equation 6, measures the proportion where the user had a correct label that was subsequently misclassified by the model. Aligned with that, $R_{A_+}$ and $R_{A_-}$ in equation 7 measure positive and negative alteration rates, respectively. Hence, an effective model will have high $R_{A_+}$ and low $R_{A_-}$. We can measure user improvement by comparing if the personalised label corrections made by the model improved, did not improve, or maintained the original users' accuracy. We also measure post alteration accuracy that compares the user's accuracy before and after applying the model.

Tables 1–4 show experimental results with the evaluation defined above. Table 1 shows user improvement, table 2 displays the accuracy after alterations from HAICO-CN in comparison with the original user's accuracy, followed by tables 3 and 4 showing positive and negative alteration as computed in equation 6 and alteration rates from equation 7. The shaded rows in tables 1 and 2 contrast testing users who met the on-boarding condition (see second step in Section 3.3), against all testing

users in the unshaded rows (note: for the CIFAR10 simulation, the two sets are the same since all users met the condition). Note that tables 3 and 4 show results for on-boarded users from the shaded rows of tables 1 and 2.

## 4.4 SIMULATION RESULTS

First row of table 1 displays the number of testing users that improved (I), maintained (M) or did not improve (NI) with the use of HAICO-CN for the simulation results on CIFAR-10. Notice how the number of I users increases and NI users decreases with $K$, showcasing that more effective personalisation depends on larger $K$. For instance, at $K = 3$, table 1 shows that all 15 users improved, and table 2 displays that the average accuracy after alteration is larger than the user's original accuracy. Nevertheless, as $K$ decreases, the new accuracy decreases slightly as a result of lower number of improved users. Similarly, the simulation results in table 3 highlights the increase of $A_+$ with higher $K$, accompanied by a decline in negative alterations $A_-$. Additionally, table 4 shows an increasing alteration rate with $K$, reflecting the larger proportion of positive alterations and smaller proportion of negative alterations with an increasing value for $K$. In appendix A, figure 3 displays the three noise matrices, each used for simulating five users. Figure 4 illustrates the estimated noise matrices for each cluster with $K \in \{1, 2, 3\}$. Notably, the noise matrices resulted for $K = 3$ closely resemble those used in the synthesis of 15 simulated users.

## 4.5 PUBLIC DATASET RESULTS

According to table 1, all on-boarded users in every experiment have improved their accuracy with HAICO-CN. Even considering all users, the method tends to improve the performance of the majority of users. Similarly to the simulated case, the number of improved users increases with $K$. Table 2 shows that the accuracy after alterations for the on-boarded users in CIFAR-10N, CIFAR-10H, Fashion-MNIST-H and Chaoyang increase by approximately 15%, 5%, 21%, 6%, respectively. Table 3 shows that negative alterations for on-boarded users tend to decrease as $K > 1$. On CIFAR-10N and Fashion-MNIST-H positive alterations increase with $K$, but CIFAR-10H and Chaoyang show the opposite trend. Nevertheless, the accuracy for all datasets increases as a function of $K$, as shown in table 2 because of the declining negative alterations. A detailed class-wise breakdown of alterations for each experiment is presented in appendix C. Table 4 shows that HAICO-CN has high positive alteration rates compared to negative alteration rates.

| Dataset | Users | K=1 | | | K=2 | | | K=3 | | |
|---|---|---|---|---|---|---|---|---|---|---|
| | | I | M | NI | I | M | NI | I | M | NI |
| With simulated data | | | | | | | | | | |
| CIFAR-10 | 15 | 5 | 0 | 10 | 9 | 0 | 6 | 15 | 0 | 0 |
| | 15 | 5 | 0 | 10 | 9 | 0 | 6 | 15 | 0 | 0 |
| With real-world data | | | | | | | | | | |
| CIFAR-10N | 80 | 80 | 0 | 0 | 80 | 0 | 0 | 80 | 0 | 0 |
| | 80 | 80 | 0 | 0 | 80 | 0 | 0 | 80 | 0 | 0 |
| CIFAR-10H | 2571 | 2548 | 0 | 23 | 2566 | 1 | 4 | 2567 | 1 | 3 |
| | 2022 | 2022 | 0 | 0 | 2022 | 0 | 0 | 2022 | 0 | 0 |
| Fashion-MNIST-H | 183 | 182 | 0 | 1 | 183 | 0 | 0 | 183 | 0 | 0 |
| | 182 | 182 | 0 | 0 | 182 | 0 | 0 | 182 | 0 | 0 |
| Chaoyang | 3 | 2 | 0 | 1 | 2 | 0 | 1 | 3 | 0 | 0 |
| | 2 | 2 | 0 | 0 | 2 | 0 | 0 | 2 | 0 | 0 |

Table 1: Number of users who improved (I), maintained (M) and did not improve (NI).

| Dataset | User's Accuracy | Accuracy After Alterations | | |
|---|---|---|---|---|
| | | K=1 | K=2 | K=3 |
| With simulated data | | | | |
| CIFAR-10 | 0.84001 | 0.83478 | 0.84500 | 0.87875 |
| | 0.84001 | 0.83478 | 0.84500 | 0.87875 |
| With real-world data | | | | |
| CIFAR-10N | 0.83648 | 0.98775 | 0.98913 | 0.98915 |
| | 0.83648 | 0.98775 | 0.98913 | 0.98915 |
| CIFAR-10H | 0.94873 | 0.99184 | 0.99304 | 0.99318 |
| | 0.93999 | 0.99143 | 0.99260 | 0.99277 |
| Fashion-MNIST-H | 0.67226 | 0.86483 | 0.87849 | 0.87693 |
| | 0.66249 | 0.86432 | 0.87786 | 0.87636 |
| Chaoyang | 0.90270 | 0.91937 | 0.94123 | 0.94657 |
| | 0.85818 | 0.91500 | 0.92714 | 0.92374 |

Table 2: Initial accuracy vs the accuracy after alterations.

| Dataset | K=1 | | K=2 | | K=3 | |
|---|---|---|---|---|---|---|
| | $A_+$ | $A_-$ | $A_+$ | $A_-$ | $A_+$ | $A_-$ |
| With simulated data | | | | | | |
| CIFAR-10 | 0.81471 | 0.16138 | 0.83784 | 0.15363 | 0.94372 | 0.13362 |
| With real-world data | | | | | | |
| CIFAR-10N | 0.95281 | 0.00545 | 0.95409 | 0.00404 | 0.95421 | 0.00403 |
| CIFAR-10H | 0.94196 | 0.00552 | 0.93886 | 0.00412 | 0.94191 | 0.00412 |
| FashionM-H | 0.73524 | 0.08139 | 0.75808 | 0.07305 | 0.75438 | 0.07430 |
| Chaoyang | 0.79429 | 0.06476 | 0.68619 | 0.03284 | 0.73769 | 0.04525 |

Table 3: Positive and negative alterations.

| Dataset | K=1 | | K=2 | | K=3 | |
|---|---|---|---|---|---|---|
| | $R_{A_+}$ | $R_{A_-}$ | $R_{A_+}$ | $R_{A_-}$ | $R_{A_+}$ | $R_{A_-}$ |
| With simulated data | | | | | | |
| CIFAR-10 | 0.83467 | 0.16533 | 0.84505 | 0.15495 | 0.87597 | 0.12403 |
| With real-world data | | | | | | |
| CIFAR-10N | 0.99431 | 0.00569 | 0.99578 | 0.00422 | 0.99579 | 0.00421 |
| CIFAR-10H | 0.99417 | 0.00583 | 0.99563 | 0.00437 | 0.99564 | 0.00436 |
| FashionM-H | 0.90033 | 0.09967 | 0.91211 | 0.08789 | 0.91034 | 0.08966 |
| Chaoyang | 0.92461 | 0.07539 | 0.95433 | 0.04567 | 0.94221 | 0.05779 |

Table 4: Alteration rates.

In Table 6 in appendix D, accuracy comparisons show the whole model $m_{\theta_k}(.)$ outperforming the base model $f_{\psi_k}(.)$, with the OVA SVM exhibiting high accuracy. Table 7 in appendix E highlights

effective joint decision-making, even when both human and base model are incorrect, showcasing the capacity to learn joint biases.

## 4.6 ABLATION STUDY

While we report the main ablation study conclusions here, tables and figures are in the appendices. Starting from appendix F, the experiment with $K \in \{1, 2, 3, 6, 10\}$ confirms that increasing $K$ improves users' post alteration accuracy until $K = 3$, which then decreases for $K > 3$ because of the reduction of the number of users per cluster. The ablation with $G \in \{1, 3, 5\}$ in appendix G shows that larger values of $G$ are positively correlated with accuracy. The ablation with different backbone models were conducted with DenseNet-121 and ResNet-50 in addition to ViT/B-16 in appendix I, where results in table 11 show that all testing users are improved in all 3 cases. Furthering the same ablation, we compare the results to methods from the literature and confirm our superior performance irrespective of the backbone model in table 12. Appendix J evaluates the effect of $\lambda$ in the loss function, where results show that higher post alteration accuracy is achieved with $\lambda = 0.1$.

## 5 COMPARATIVE ANALYSIS

In table 5, we compare our results with the following competing methods proposed in literature: confidence comparison (Raghu et al., 2019), cross-entropy surrogate (Mozannar & Sontag, 2020), One-Vs-All Verma & Nalisnick (2022), differentiable triage (Okati et al., 2021), mixture-of-experts (Madras et al., 2018) and realizable surrogate (Mozannar et al., 2023) using the implementations released by Mozannar et al. (2023) and data preparation details are in appendix K. The accuracy comparison against ground truth annotations on the test set (table 5) involves related methods trained with and

| Method | CIFAR-10N | CIFAR-10H | FashionM-H | Chaoyang |
|---|---|---|---|---|
| | Trained with Ground Truth | | | |
| Madras et al. (2018) | 0.83070 | 0.81200 | 0.60015 | 0.58345 |
| Raghu et al. (2019) | 0.97030 | 0.97090 | 0.80053 | 0.86255 |
| Mozannar & Sontag (2020) | 0.94890 | 0.96690 | 0.72948 | 0.70593 |
| Okati et al. (2021) | 0.94020 | 0.94390 | 0.70400 | 0.76484 |
| Verma & Nalisnick (2022) | 0.95880 | 0.97410 | 0.79375 | 0.84478 |
| Mozannar et al. (2023) | 0.94790 | 0.97570 | 0.77533 | 0.87237 |
| | Trained without Ground Truth | | | |
| Madras et al. (2018) | 0.86050 | 0.88380 | 0.59975 | 0.59513 |
| Raghu et al. (2019) | 0.96680 | 0.96880 | 0.78335 | 0.86208 |
| Mozannar & Sontag (2020) | 0.92540 | 0.96880 | 0.74910 | 0.67741 |
| Okati et al. (2021) | 0.88110 | 0.90020 | 0.75220 | 0.71949 |
| Verma & Nalisnick (2022) | 0.94500 | 0.97110 | 0.60901 | 0.86676 |
| Mozannar et al. (2023) | 0.94460 | 0.96820 | 0.75150 | 0.86676 |
| **Ours (k=3)** | **0.98915** | **0.99277** | **0.87636** | **0.92374** |

Table 5: Comparative analysis against the proposed methods in the literature

without ground truth (top and bottom parts). Remarkably, our ground truth free trained models outperforms those trained with ground truth.

## 6 LIMITATION AND FUTURE WORK

HAICO-CN acknowledges the potential limitation of assuming distinct noise patterns across multiple user groups, especially when users exhibit comparable noisy patterns. Exploring automated methods to determine the optimal number of clusters could enhance the effectiveness of HAICO-CN. Additionally, the current validation set design necessitates each testing user to provide 20 labels for each class, suggesting future research for more efficient few-shot onboarding process. It is essential to address the susceptibility of learned clusters to compromising individual privacy, emphasizing the need for future works to carefully balance effective clustering with preserving user privacy.

HAICO-CN, is focused on multi-class classification, aims to broaden its scope to address multi-label scenarios. One challenge would be estimating noise transition matrices and we seek to leveraging insights from recent literature Li et al. (2022); Kye et al. (2022).

## 7 CONCLUSION

This paper introduces HAICO-CN, a novel Human-AI collaborative method that improves joint decision-making through personalized model training using cluster-wise noisy-label augmentation. We propose new evaluation criteria and empirically demonstrate HAICO-CN's superior performance on datasets, including CIFAR-10N, CIFAR-10H, Fashion-MNIST-H, and Chaoyang histopathology, surpassing state-of-the-art human-AI collaboration methods.

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

APPENDIX

# A    SIMULATION DETAILS

This section provides some more details about the simulation experiment explained in the main paper. Figure 3 shows the 3 noise matrices used for simulating 5 users from each cluster. Following that, the figure 4 illustrates the estimated noise matrices for each cluster when $K \in \{1, 2, 3\}$. It is interesting to see that the noise matrices resulted in $K = 3$ in figure 4 are similar to the ones used for creating 15 simulated users. This confirms the effectiveness of the clustering process as it has manages to identify noise patterns of users and to cluster them accurately.

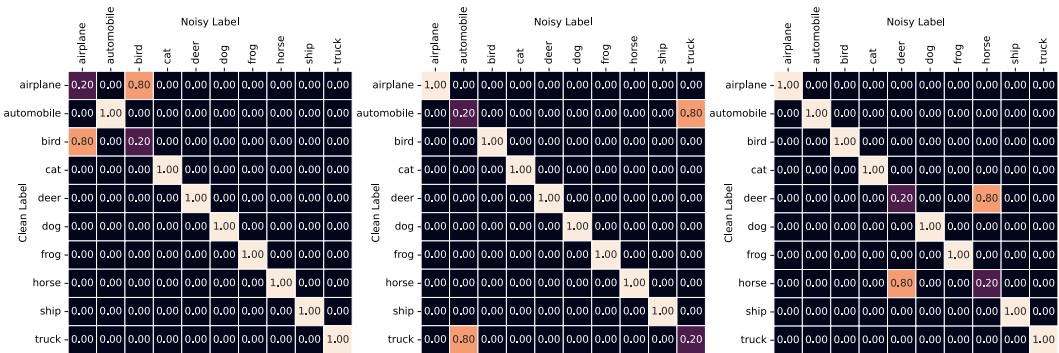

Figure 3: Noise matrices used for simulating users.

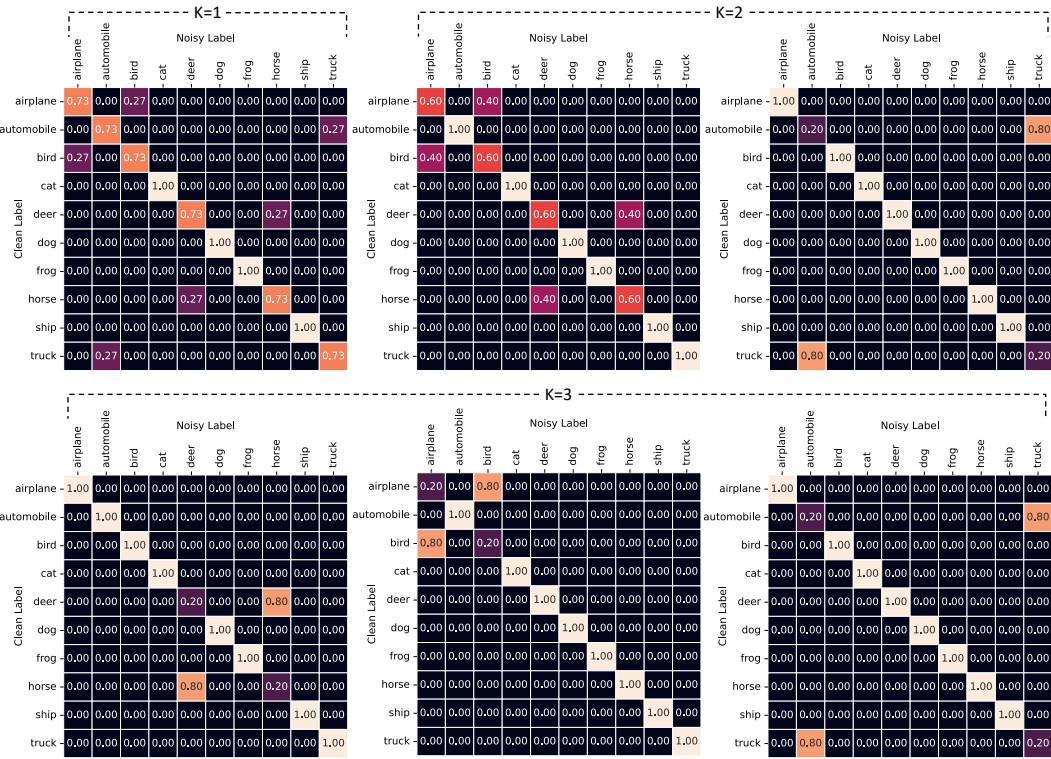

Figure 4: Estimated noise matrices for each cluster when $K \in \{1, 2, 3\}$ from the simulation.

# B  NOISE PATTERNS IN CLUSTERS

This section visualises the noise distribution in clusters after performing Fuzzy K-Means with K=3. Figures 5, 6 and 7 illustrates clusters from CIFAR-10 simulation, Fashion-MNIST-H and Chaoyang experiments. Those cluster noise visualisations are complimented with sample images where label noise was found and positively altered by the model.

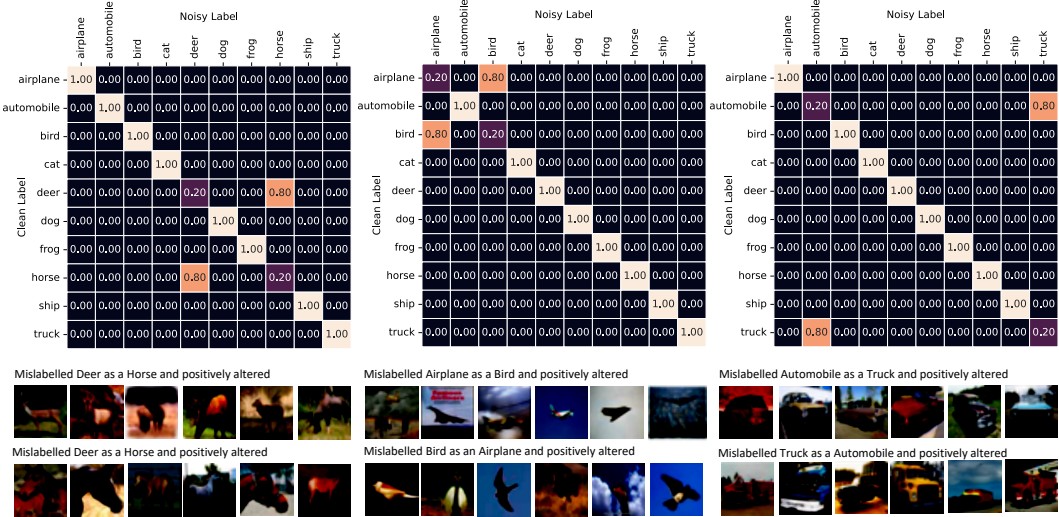

Figure 5: Noise matrices when K=3 in CIFAR-10 simulation experiment

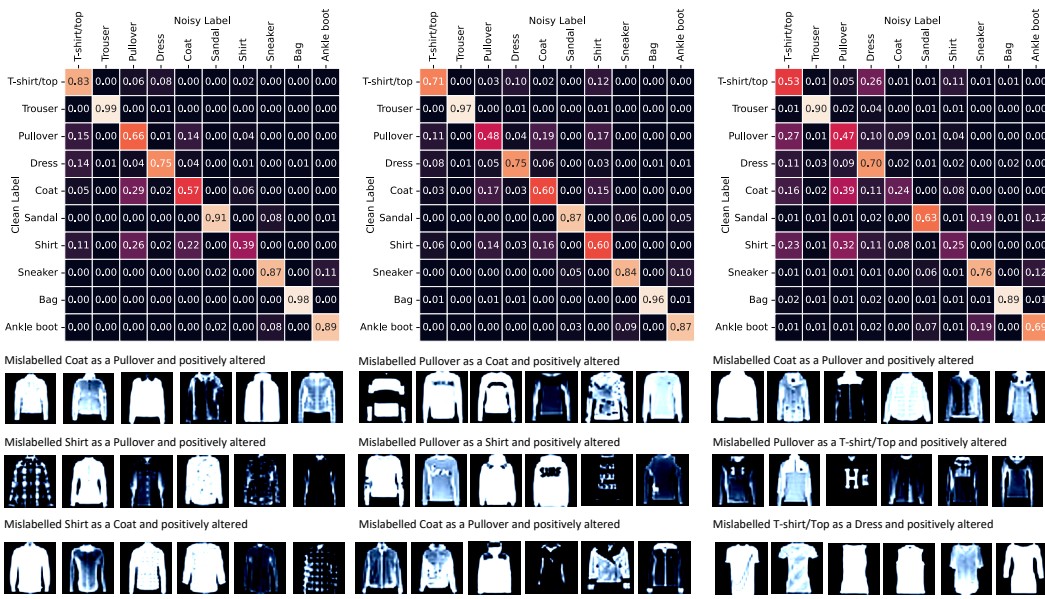

Figure 6: Noise matrices when K=3 in Fashion-MNIST-H experiment

# C  THE BREAKDOWN OF POSITIVE AND NEGATIVE ALTERATIONS

This section presents figure 8 that shows the class-wise (CIFAR-10N) number of errors made by humans (first bar – yellow) and how many out of them were positively altered by the model (second bar – orange), and the number of correct cases from humans (third bar – purple) and how many labels out of them were negatively altered by the model (fourth bar – blue). This figure 8 shows

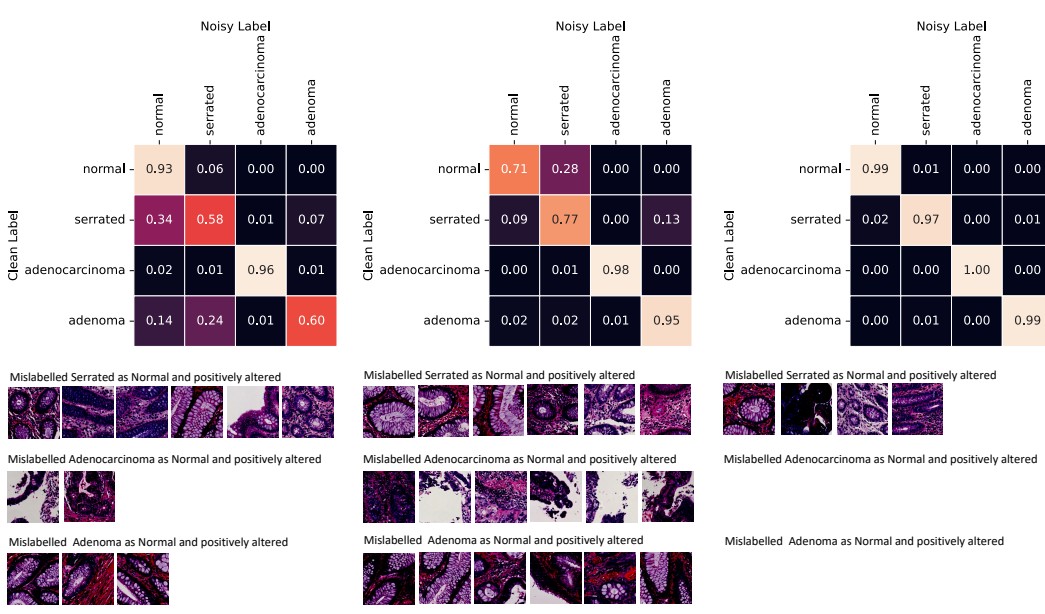

Figure 7: Noise matrices when K=3 in Chaoyang experiment

these results for different values of $K \in \{1, 2, 3\}$. Similarly, figure, 10, 11 and 9 show te same results for CIFAR-10H, Fashin-MNIST-H and Chaoyang datasets, respectively.

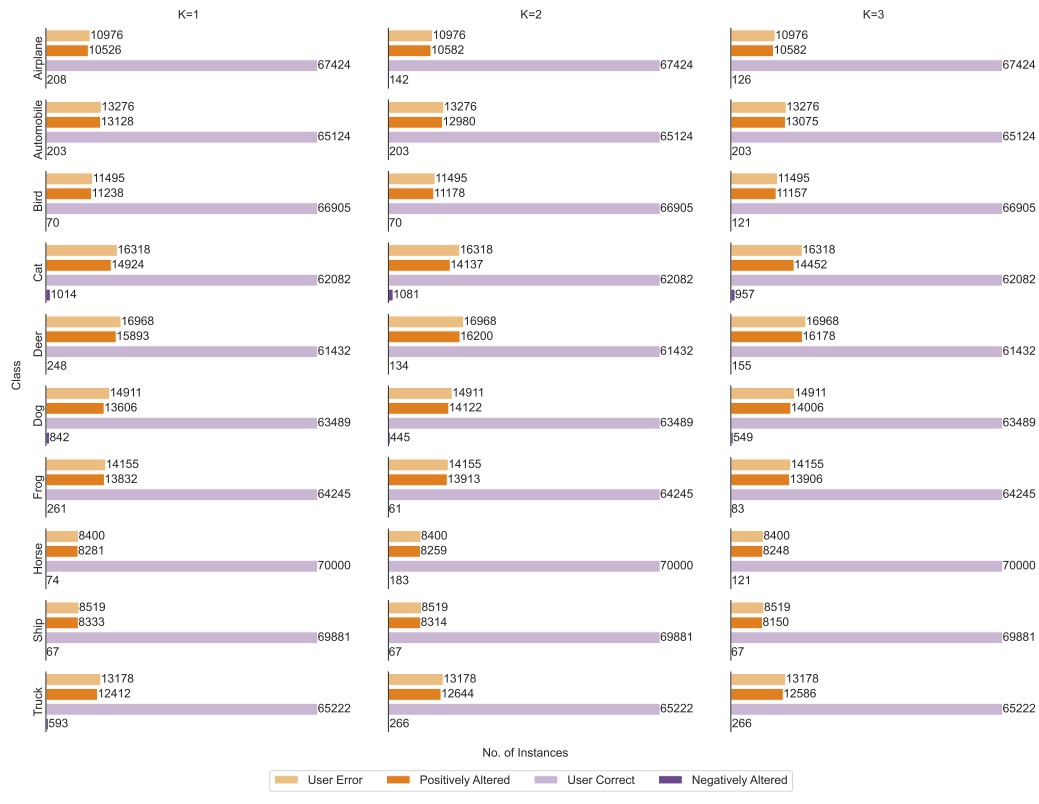

Figure 8: Alterations resulted in the experiment with CIFAR-10N.

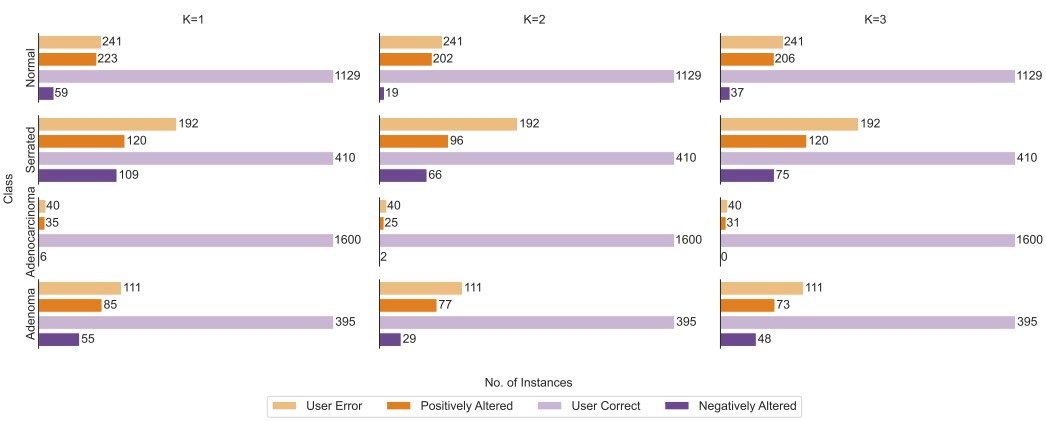

Figure 9: Alterations resulted in the experiment with Chaoyang.

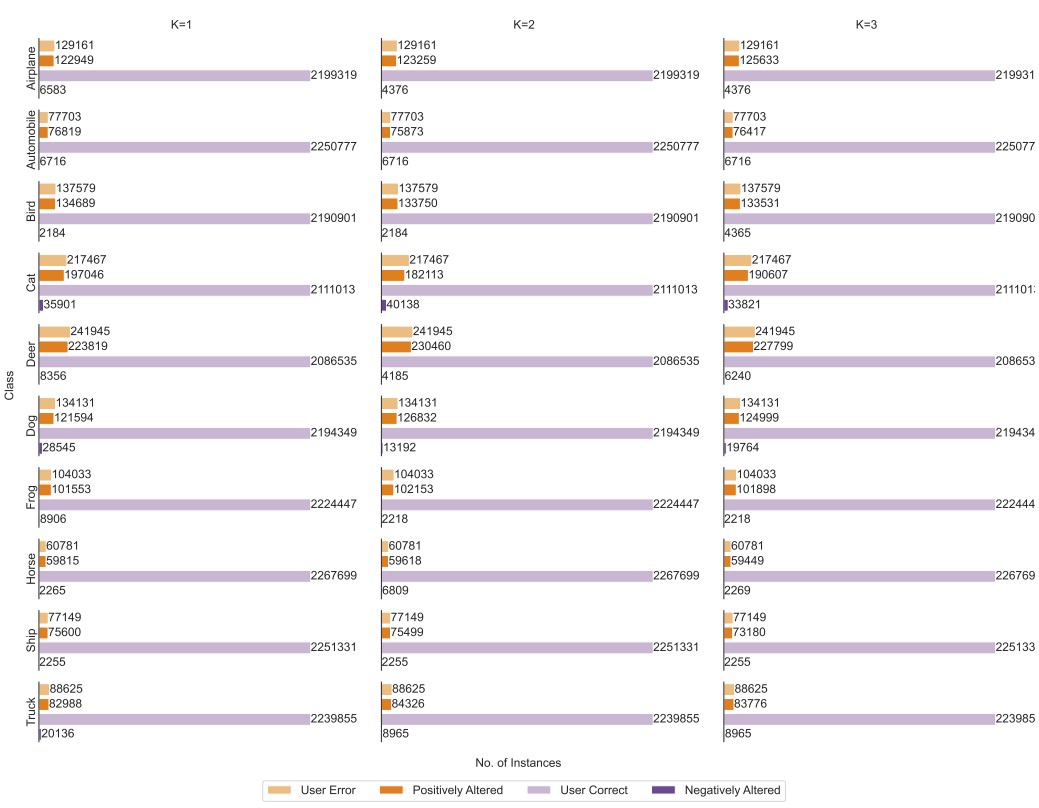

Figure 10: Alterations resulted in the experiment with CIFAR-10H.

# D    THE INTERMEDIATE RESULTS OF EXPERIMENTS

Table 6 reports the accuracy of the base and whole models trained during different steps in the experiments and how users were classified into clusters during training for different values of $K$.

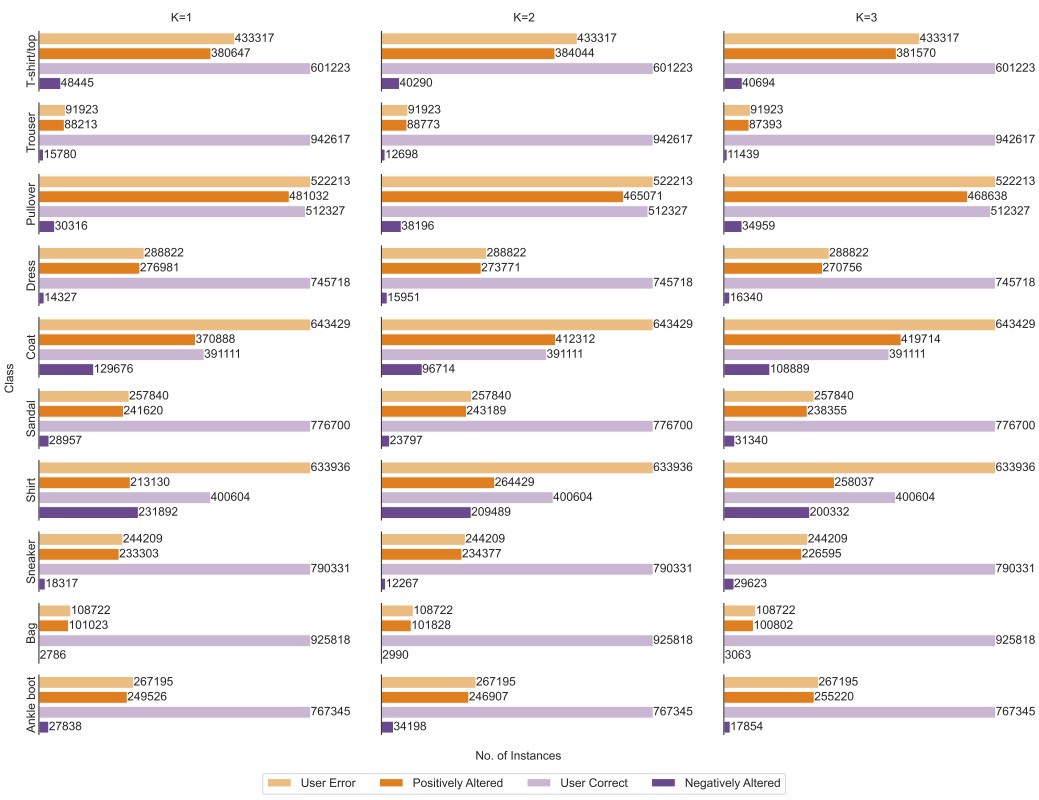

Figure 11: Alterations resulted in the experiment with Fashin-MNIST-H.

# E    DISTRIBUTION OF DECISIONS MADE BY HUMAN, AI AND HUMAN-AI COLLABORATION

The table 7 shows how decisions from human, base model and joint decisions are distributed at each experiment conducted in section 4. This proportions are computed using the testing set. Decision of human, or the AI model $f_{\psi_k}(.)$, or the collaboration $m_{\theta_k}(.)$ are divided into correct ($\checkmark$), if their label is equal to the target, or wrong ($\times$), otherwise. According to table 7, in all experiments, the majority of correct joint decisions are resulted following both correct human and AI counterparts. On the contrary, the least proportion of incorrect joint decisions are made when both individual parties are correct. Further, the results reflects the tendency of joint decision being correct than being wrong when at least one member of the Human-AI team is correct, as anticipated in a collaborative setting. These observations underscore the effectiveness of our approach for collaborative decision making.

An interesting observation is that the in small proportion, joint decision turns out to be correct while both individual counterparts are wrong. We believe this showcases the capacity of out approach to learn the joint biases posed by individual parties and intervene in cases where both are weak.

# F    PERFORMANCE AS A FUNCTION OF $K$

The effect of having more clusters was studied by extending the experiment done with CIFAR-10N dataset with VIT/B-16 base model. The results in table 8 indicate that from $K = 1$ to $K = 3$, the accuracy increases and, for $K > 3$, it starts to decrease. Even though all testing users had their accuracy improved in all experiments, their accuracy gain has been slightly impacted by $K$. This demonstrates that having larger $K$ does not guarantee the best accuracy gain. Hence, as $K$ increases, the number of users per cluster during training decreases. This means that the augmented

| Dataset | $f_{\psi_k}(.)$ Accuracy | K and (User Distribution in Clusters) | OVA SVM Accuracy | $m_{\theta_k}(.)$ Accuracy |
|---|---|---|---|---|
| | | With Simulated Data | | |
| CIFAR-10 | 0.85410 | 1: (15) | - | 0.85176 |
| | | 2: (10, 5) | 0.99999 | 0.86833 / 0.86066 |
| | | 3: (5, 5, 5) | 0.99999 | 0.87710 / 0.87706 / 0.88253 |
| | | With Public Datasets | | |
| CIFAR-10N CIFAR-10H | 0.98150 | 1: (79) | - | 0.98850 |
| | | 2: (31, 48) | 0.74750 | 0.98996 / 0.98886 |
| | | 3: (32, 27, 20) | 0.60500 | 0.98926 / 0.98976 / 0.98930 |
| Fashion-MNIST-H | 0.86303 | 1: (183) | - | 0.86599 |
| | | 2: (90, 93) | 0.99999 | 0.87239 / 0.88483 |
| | | 3: (35, 71, 77) | 0.99999 | 0.88345 / 0.87972 / 0.87377 |
| Chaoyang | 0.88639 | 1: (3) | - | 0.91771 |
| | | 2: (1, 2) | 0.99999 | 0.91413 / 0.95075 |
| | | 3: (1, 1, 1) | 0.99400 | 0.92535 / 0.91678 / 0.99174 |

Table 6: Accuracy of the base model $f_{\psi_k}(.)$ and whole model $m_{\theta_k}(.)$ for $K \in \{1, 2, 3\}$ (where we show the number of users in each cluster).

| Human | AI $f_{\psi_k}(.)$ | Collabora-tion $m_{\theta_k}(.)$ | CIFAR10-N K=1 | K=2 | K=3 | CIFAR10-H K=1 | K=2 | K=3 | Fashion-MNIST-H K=1 | K=2 | K=3 | Chaoyang K=1 | K=2 | K=3 |
|---|---|---|---|---|---|---|---|---|---|---|---|---|---|---|
| ✗ | ✓ | ✓ | 0.05147 | 0.05153 | 0.05156 | 0.05603 | 0.05595 | 0.05614 | 0.04542 | 0.04469 | 0.04191 | 0.03667 | 0.03205 | 0.03351 |
| ✓ | ✗ | ✓ | 0.00530 | 0.00651 | 0.00632 | 0.01816 | 0.02261 | 0.02156 | 0.11702 | 0.15052 | 0.13313 | 0.01125 | 0.02445 | 0.01821 |
| ✓ | ✓ | ✓ | 0.93867 | 0.93786 | 0.93805 | 0.91665 | 0.91351 | 0.91456 | 0.75281 | 0.72130 | 0.73844 | 0.92294 | 0.91889 | 0.92156 |
| ✗ | ✗ | ✓ | 0.00048 | 0.00048 | 0.00045 | 0.00060 | 0.00054 | 0.00051 | 0.03953 | 0.04278 | 0.04531 | 0.00081 | 0.00032 | 0.00130 |
| ✗ | ✓ | ✗ | 0.00151 | 0.00127 | 0.00130 | 0.00194 | 0.00186 | 0.00182 | 0.00139 | 0.00332 | 0.00416 | 0.00202 | 0.00712 | 0.00486 |
| ✓ | ✗ | ✗ | 0.00147 | 0.00113 | 0.00112 | 0.00501 | 0.00388 | 0.00388 | 0.01697 | 0.01367 | 0.01356 | 0.01854 | 0.00939 | 0.01295 |
| ✓ | ✓ | ✗ | 0.00005 | 0.00000 | 0.00001 | 0.00019 | 0.00000 | 0.00000 | 0.00070 | 0.00201 | 0.00237 | 0.00000 | 0.00000 | 0.00000 |
| ✗ | ✗ | ✗ | 0.00105 | 0.00123 | 0.00119 | 0.00143 | 0.00166 | 0.00153 | 0.02617 | 0.02171 | 0.02111 | 0.00777 | 0.00777 | 0.00761 |

Table 7: Proportion that each combination of Human, AI, or Collaboration is correct (✓) or incorrect (✗). Columns sum to 1 to indicate all possible combinations, and we show results for varying values of $K$.

noisy labels may over personalise biases which may lead to a less generalisable model for testing users.

| $K$ | Accuracy | A+ | A- | RA+ | RA- |
|---|---|---|---|---|---|
| K=1 | 0.98775 | 0.95281 | 0.00545 | 0.99431 | 0.0057 |
| K=2 | 0.98913 | 0.95409 | 0.00404 | 0.99578 | 0.0042 |
| K=3 | 0.98915 | 0.95421 | 0.00403 | 0.99579 | 0.0042 |
| K=6 | 0.98775 | 0.94382 | 0.00367 | 0.99613 | 0.0039 |
| K=10 | 0.98275 | 0.91350 | 0.00378 | 0.99588 | 0.0041 |

Table 8: Performance on CIFAR-10N as a function of the number of clusters $K$.

## G   PERFORMANCE AS A FUNCTION OF NOISY LABEL AUGMENTATION $G$

The effect of the number of times $G$ that noisy labels were augmented in cluster $\hat{\mathcal{D}}_k$ is explored by extending the CIFAR-10N experiment with VIT/B-16. The results in 9 shows that larger $G$ promotes a slight increase in the users' post alteration accuracy. Note that $K$ was fixed at 3 for this experiment.

| $G$ | Accuracy | A+ | A- | RA+ | RA- |
|---|---|---|---|---|---|
| 1 | 0.98897 | 0.95300 | 0.00402 | 0.99580 | 0.00420 |
| 3 | 0.98915 | 0.95421 | 0.00403 | 0.99579 | 0.00421 |
| 5 | 0.98922 | 0.95221 | 0.00354 | 0.99630 | 0.00370 |

Table 9: Performance on CIFAR-10N as a function of the noisy label augmentation hyper-parameter $G$.

| Asymmetric Noise Rate | User's Accuracy | Post alteration accuracy (K=3) |
|---|---|---|
| 40% | 0.91984 | 0.99232 |
| 60% | 0.88001 | 0.96775 |
| 80% | 0.84001 | 0.87875 |
| 90% | 0.82022 | 0.86841 |

Table 10: Performance on CIFAR-10 as a function of noise rate

## H   PERFORMANCE AS A FUNCTION OF NOISE RATE

The robustness of the approach for different noise rates were studied by extending the simulation with CIFAR-10 to different noise rates. The obtained results are reported in the table 10. A ResNet-18 pretrained on ImageNet-1K was used as the backbone for base model. The same simulation data preparation process in section 4.2 was followed here.

## I   THE ABLATION WITH DIFFERENT BACKBONE MODELS AS BASE MODEL

This experiment tests different backbones as the base model for the experiment on CIFAR-10N dataset. The CIFAR-10N experiment was done in section 4.2 with a VIT/B-16 as the base model $f_{\psi_k}(.)$, and here we use DenseNet-121 and Resnet-50 as $f_{\psi_k}(.)$.

The results in table 11 showcases that different base models improve users in different degrees as accuracy after alterations is different among them. Yet, it consistently surpasses the original accuracy of users and all the on-boarded users were improved irrespectively of the base model.

It is important to emphasise that as the $f_{\psi_k}(.)$ changes, the consensus estimation in section 3.1 changes. Following that, the number of users chosen for labelling at least 20 images from each class varies. This also changes the number of users in the test set and the recorded original accuracy in the table 11. To be specific, the experiments with ResNet-50 and DenseNet-121 were conducted respectively with 155 and 157 users identified for labelling 20 images per class. In the experiment with ResNet-50, 77 were in the training set and 78 were in the testing set. In the case with DenseNet-121, it was 78 and 79 in training and testing sets, respectively. The recorded results and user distribution for the experiment with ViT/B-16 are same as in the main paper.

| Backbone Model | K | Original Accuracy | Accuracy After Alterations | A+ | A- | RA+ | RA- |
|---|---|---|---|---|---|---|---|
| ResNet-50 | K=1 | | 0.96593 | 0.86007 | 0.01484 | 0.98304 | 0.01696 |
| | K=2 | 0.84614 | 0.96770 | 0.86226 | 0.01314 | 0.98499 | 0.01501 |
| | K=3 | | 0.96748 | 0.86303 | 0.01352 | 0.98458 | 0.01542 |
| DenseNet-121 | K=1 | | 0.96870 | 0.85504 | 0.01066 | 0.98769 | 0.01231 |
| | K=2 | 0.84644 | 0.96859 | 0.85350 | 0.01053 | 0.98781 | 0.01219 |
| | K=3 | | 0.96787 | 0.85647 | 0.01190 | 0.98630 | 0.01370 |
| Vit/B-16 | K=1 | | 0.98775 | 0.95281 | 0.00545 | 0.99431 | 0.00569 |
| | K=2 | 0.83648 | 0.98913 | 0.95409 | 0.00404 | 0.99578 | 0.00422 |
| | K=3 | | 0.98915 | 0.95421 | 0.00403 | 0.99579 | 0.00421 |

Table 11: Ablation with CIFAR-10N using backbone models as the base model $f_{\psi_k}(.)$.

Further, we extend the comparative analysis in section 5 and use the two backbones with methods from literature to examine the performance. From the results in the table 12, our approach consistently outperforms the methods in literature.

| Method | ResNet50 | DenseNet121 | ViTB16 |
|---|---|---|---|
| | | With Ground Truth | |
| Madras et al. (2018) | 0.85080 | 0.84120 | 0.83070 |
| Raghu et al. (2019) | 0.87070 | 0.82810 | 0.97030 |
| Mozannar & Sontag (2020) | 0.85140 | 0.85020 | 0.94890 |
| Okati et al. (2021) | 0.81030 | 0.80210 | 0.94020 |
| Verma & Nalisnick (2022) | 0.70080 | 0.63320 | 0.95880 |
| Mozannar et al. (2023) | 0.78220 | 0.84960 | 0.94790 |
| | | Without Ground Truth | |
| Madras et al. (2018) | 0.84270 | 0.84740 | 0.86050 |
| Raghu et al. (2019) | 0.83160 | 0.87880 | 0.96680 |
| Mozannar & Sontag (2020) | 0.70300 | 0.84890 | 0.92540 |
| Okati et al. (2021) | 0.80030 | 0.70550 | 0.88110 |
| Verma & Nalisnick (2022) | 0.62410 | 0.59320 | 0.94500 |
| Mozannar et al. (2023) | 0.65880 | 0.84700 | 0.94460 |
| **Ours (k=1)** | **0.96593** | **0.96870** | **0.98775** |
| **Ours (k=2)** | **0.96770** | **0.96859** | **0.98913** |
| **Ours (k=3)** | **0.96748** | **0.96787** | **0.98915** |

Table 12: Comparison between HAICO-CN and competing methods in the literature with different base models using CIFAR-10N.

| $K$ | $\lambda = 0$ | $\lambda = 0.01$ | $\lambda = 0.1$ | $\lambda = 1$ | $\lambda = 10$ |
|---|---|---|---|---|---|
| | | ResNet-50 [Test accuracy - 0.91140] | | | |
| K=1 | 0.92437 | 0.94271 | 0.96593 | 0.93427 | 0.91874 |
| K=2 | 0.92945 | 0.94373 | 0.96770 | 0.93986 | 0.92910 |
| K=3 | 0.92530 | 0.93466 | 0.96747 | 0.93227 | 0.92991 |
| | | DenseNet-121 [Test accuracy - 0.92290] | | | |
| K=1 | 0.93685 | 0.94787 | 0.96870 | 0.94979 | 0.92103 |
| K=2 | 0.93635 | 0.95009 | 0.96859 | 0.93733 | 0.93062 |
| K=3 | 0.94297 | 0.94808 | 0.96787 | 0.94311 | 0.93184 |
| | | ViT-B/16 [Test accuracy - 0.98150] | | | |
| K=1 | 0.97219 | 0.97267 | 0.98775 | 0.97985 | 0.97347 |
| K=2 | 0.98213 | 0.98149 | 0.98913 | 0.97588 | 0.96950 |
| K=3 | 0.97751 | 0.98219 | 0.98915 | 0.97899 | 0.97306 |

Table 13: Post alteration accuracy variation in terms of $\lambda$ that weights the second term of the loss in equation 5 (with CIFAR-10N).

## J  TESTING $\lambda$ IN THE LOSS FUNCTION

Here, we study how the second term in the loss function in equation 5 affects the post alteration accuracy. We conduct a range of experiments with $\lambda \in \{0, 0.01, 0.1, 1, 10\}$. Using CIFAR-10N dataset, three sets of experiments were conducted using ResNet-50, DenseNet-121 and Bit/B-16 as base models. Even though all users were improved in every experiment, the results in table 13 show how post alteration accuracy vary with $\lambda$. It is clear that the highest post alteration accuracy is centered around $\lambda = 0.1$ for all 3 base models.

## K  DATA PREPARATION FOR COMPARATIVE ANALYSIS

The comparison is performed by training the methods from literature using (with) and not using (without) the ground truth in two sets of experiments. The training set consensus $\bar{\mathbf{y}}$ was used when training without the ground truth.

| Method | Noise Rate | | |
|---|---|---|---|
| | 10% | 30% | 40% |
| CE | 0.888 | 0.817 | 0.761 |
| LDMI Xu et al. (2019) | 0.911 | 0.912 | 0.840 |
| M-Up Zhang et al. (2017) | 0.933 | 0.833 | 0.777 |
| JPL Kim et al. (2021) | 0.942 | 0.925 | 0.907 |
| PCIL Yi & Wu (2019) | 0.931 | 0.929 | 0.916 |
| Dmix Li et al. (2020) | 0.938 | 0.925 | 0.917 |
| ELR Liu et al. (2020) | 0.954 | 0.947 | 0.930 |
| MOIT Ortego et al. (2021) | 0.942 | 0.941 | 0.932 |
| C2D Zheltonozhskii et al. (2022) | - | - | 0.937 |
| UNICON Karim et al. (2022) | 0.953 | 0.948 | 0.941 |
| Ours (K=3) | 0.99784 | 0.99598 | 0.99273 |

Table 14: Comparison with proposed noisy label learning methods in literature. Accuracy for those methods were referenced from Karim et al. (2022) and Zheltonozhskii et al. (2022).

| K | Silhouette Score |
|---|---|
| 2 | 0.34752 |
| 3 | 0.55195 |
| 4 | 0.37054 |
| 5 | 0.18677 |
| 6 | 0.00565 |
| 7 | 0.00637 |
| 8 | 0.00194 |
| 9 | 0.00469 |
| 10 | 0.00281 |

Table 15: Silhouette score variation as a function of K for CIFAR-10 simulation

## L  COMPARING WITH NOISY LABEL LEARNING METHODS

While recognizing the difference of HAICO-CN compared to noisy label learning techniques, we conducted a comparative experiment following the setup in Karim et al. (2022). Specifically, we simulated five users, each introducing a 10% asymmetric noise in three class pairs (Airplane-Bird, Truck-Automobile, and Horse-Deer). Subsequently, we trained and evaluated HAICO-CN with k=3. The same experiment was repeated for 30% and 40% noise rates and results are reported in table 14. A Vit-Base-16 pre-trained on ImageNet-21K has been used as the backbone for base model.

Recent noisy labeling methods such as Zheltonozhskii et al. (2022); Karim et al. (2022) have demonstrated impressive results across various levels of noise rate. Nevertheless, these approaches would require substantial modifications to effectively manage multi-rater labels or be adapted for use in a human-AI collaborative context, where models take both human and AI inputs.

## M  FUNDAMENTAL WAY TO APPROXIMATE K FOR FUZZY K-MEANS

One of the possible ways is to utilize a quantitative assessment of cluster quality, selecting the K that produces the highest score using the training users. The table 15 presents the silhouette scores for K values ranging from 2 to 10 in the simulation experiment with CIFAR10. Notably, the case with K=3 exhibits the highest score, aligning with the visual quality measure for the specified rationale in figure 4.

Another way would be to plot post alteration accuracy against K and determine K that yields highest accuracy. The figure 12 illustrates such elbow plot for CIFAR-10N experiment and K=3 has the highest accuracy.

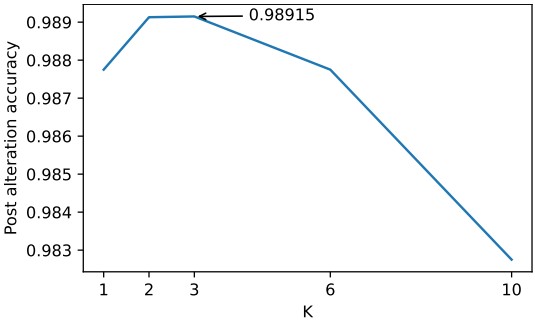

Figure 12: Elbow plot for the experiment with CIFAR-10N dataset

# N    ATTEMPT TO MODEL INTERPRETABILITY

We conducted an experiment by replacing the decision model in HAICO-CN with a decision tree model to enable interpretability. The decision tree was trained by concatenating the output logits from base model and human embedding for the training set as in the section 3.2.3.

Experiment was done for K=3 in simulation experiment with CIFAR-10 and trained decision trees are plot in the figures 13 and 14. It can be seen the decision tree uses the base model's output features (with the prefix 'b_') as a decision factor when there is user noise present in a specific class. Otherwise the tree relies on human input features with the prefix 'u_'.

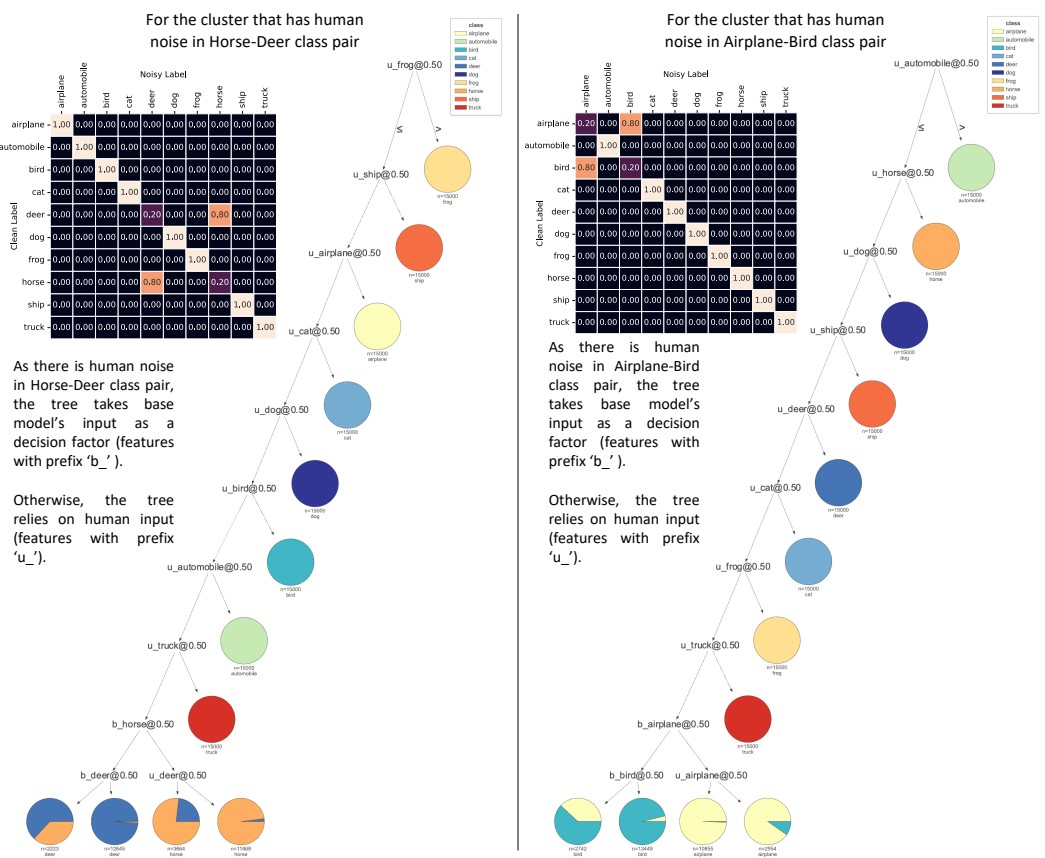

Figure 13: Decision tree behaviour when it is trained on cluster with human noise in Horse-Deer class pair (left) and Airplane-Bird class pair (right).

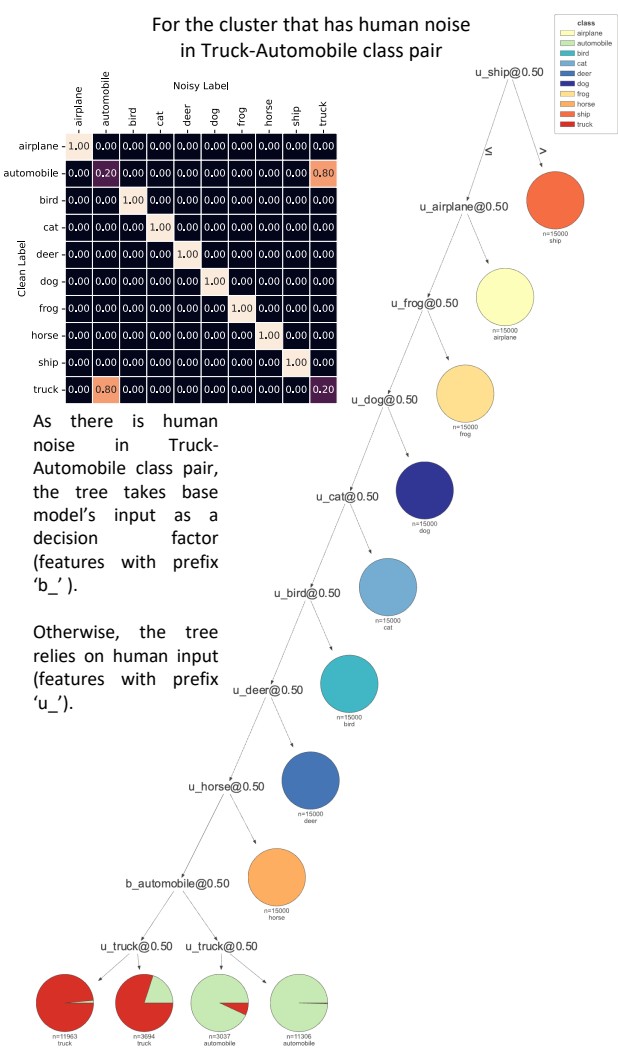

Figure 14: Decision tree behaviour when it is trained on cluster with human noise in Truck-Automobile class pair.

