# OpenReview forum: "HAICO-CN: Human-AI Collaboration By Cluster-wise Noisy-Label Augmentation"
_ICLR.cc/2024/Conference — Submitted to ICLR 2024_

### Official Review · Reviewer_bbw3 · 2023-10-29

**Soundness:** 3 good
**Presentation:** 3 good
**Contribution:** 3 good
**Rating:** 5
**Confidence:** 4

**Summary:**

- To address this gap between the AI models and humans, the authors propose a human-AI collaborative method referred to as HAICO-CN that enhances human-AI joint decision-making by training personalized models using a novel cluster-wise noisy-label augmentation technique.
- During training, HAICO-CN first identifies and clusters noise label patterns within the multi-rater data sets, followed by a cluster-wise noisy-label augmentation method that generates enough data to train a collaborative human-AI model for each cluster.
- During inference, the user follows an onboarding process, allowing HAICO-CN to select a cluster-wise human-AI model based on the user’s noisy label patterns, thereby enhancing human-AI joint decision-making performance.
- The author also proposes new evaluation criteria for assessing human-AI collaborative methods and empirically evaluate HAICO-CN across diverse datasets to validate its effectiveness.

**Strengths:**

(+) The proposed HAICO-CN is a human-AI collaborative ensemble method that enhances human-AI joint decision-making by training personalized models using a novel cluster-wise noisy-label augmentation technique.

**Weaknesses:**

- (-) The proposed method uses a noisy-label augmentation technique. However, there are no interpretations of noise level and its performances.
- (-) The model could work on user clusters that are sensitive to user bias. The user cluster selection would lead to leaking group information.

**Questions:**

- How about the cluster-wise final performances? Some cluster-wise performance plots would help understand the HAICO-CN and the role of clusters.
- What makes the model robust to noise? To show the robustness, additional interpretation would be needed using input perturbation.
- The performances seem to depend on cluster bias—an additional interpretation of clusters such as an ensemble of clusters and individual cluster-wise performances.

**Details Of Ethics Concerns:**

None.

---

> ### Author Response · Authors · 2023-11-23
> **Reply to reviewer bbw3**
>
> We sincerely appreciate the time and effort you dedicated to reviewing our paper. Below, you will find the responses and explanations for the questions raised during the review process.
>
> **How about the cluster-wise final performances? Some cluster-wise performance plots would help understand the HAICO-CN and the role of clusters.**
>
> >We thank the reviewer's suggestion on showing cluster-wise accuracy. The table below shows the cluster-wise accuracy when K=3 for the three public dataset experiments in the paper.
> | Dataset        | Post alteration accuracy (K=3) |
> | :------------- | :----------------------------: |
> | CIFAR10-N      | 0\.98903                       |
> |                | 0\.98925                       |
> |                | 0\.98929                       |
> | Fashin-MNIST-H | 0\.88151                       |
> |                | 0\.87978                       |
> |                | 0\.87094                       |
> | Chaoyang       | 0\.92763                       |
> |                | 0\.91986                       |
> |                | 0\.99222                       |
>
> >To further assist readers in understanding the role of clusters, in addition to the Figure 4 in the paper, we will also add additional visualization plots into the paper as follows:
>
> >* K=3 in CIFAR10 simulation: https://ibb.co/q7cFqBp
> >* K=3 in Chaoyang experiment: https://ibb.co/HNRshgD
> >* K=3 in Fashion-MNIST-H experiment: https://ibb.co/6B01YXr
>
> ---
>
> **What makes the model robust to noise? To show the robustness, additional interpretation would be needed using input perturbation.**
>
> >Thank you for the excellent question. While we lack a formal proof, we believe the model's robustness benefits from the complementary roles played by both the AI model and the human. Other learning-to-complement works , such as Steyvers et al. (2022) and Wilder et al. (2021), have demonstrated improved performance through a similar approach.
>
> >To empirically demonstrate the model's resilience to diverse noise levels, we conducted an ablation study with varying asymmetric noise rates. The results, including the noise rate, the simulated user's original accuracy, and the post-alteration accuracy for K=3 are presented below.
>
> | Asymmetric Noise Rate | User's Accuracy | Post alteration accuracy (K=3) |
> | :-------------------: | :-------------: | :----------------------------: |
> | 40%                   | 0\.91984        | 0\.99232                       |
> | 60%                   | 0\.88001        | 0\.96775                       |
> | 80%                   | 0\.84001        | 0\.87875                       |
> | 90%                   | 0\.82022        | 0\.86841                       |
>
> >(We further discuss the interpretation aspect of the model in the Question 3)
>
> ---
>
> **The performances seem to depend on cluster bias—an additional interpretation of clusters such as an ensemble of clusters and individual cluster-wise performances.**
>
> >Yes, the performances indeed depend on cluster bias, and we agree that more information and interpretation regarding each individual cluster is necessary. Toward this end, we have further explored two approaches:
>
> >First, we provide the visualization of the cluster-wise noise pattern (as suggested in Q1) and some samples with the label bias for
>
> >* K=3 in CIFAR10 simulation: https://ibb.co/q7cFqBp
> >* K=3 in Chaoyang experiment: https://ibb.co/HNRshgD
> >* K=3 in Fashion-MNIST-H experiment: https://ibb.co/6B01YXr
>
> >We can clearly see that each cluster represents a distinct noisy pattern and thus could better complement the human with the similar noise.
>
> >Second, we conducted an experiment that replaced the decision model in HAICO-CN with a decision tree model for interpretation: https://ibb.co/0yc2WzB
>
> >The figure in the link above shows the three decision trees for K=3 from CIFAR10 simulation using the base model’s output features (with the prefix ‘b_’) as a decision factor when there is user noise present in the classes. Otherwise the tree relies on human input features (with the prefix ‘u_’).
>
> ---
>
> >Moreover, we would also like to thank the reviewer’s insight on the potential compromise of the group anonymity given the learned cluster that might lead to the disclosure of personal information or even heighten the re-identification risk.
>
> >We propose to add the following paragraph into the limitation section of the paper:
>
> >“One limitation of the model lies in its susceptibility of the learned user clusters to compromise the privacy of individual users. Specifically, the selection of user clusters has the potential to leak sensitive group information, posing a risk to group anonymity. Future works should carefully consider the delicate balance between achieving effective clustering and preserving individual privacy”

---

> > ### Comment · Reviewer_bbw3 · 2023-11-23
> >
> > Thank you for your detailed explanations and extensive experiments. I increased my score from 3 to 5.

---

### Official Review · Reviewer_SoV5 · 2023-11-02

**Soundness:** 3 good
**Presentation:** 3 good
**Contribution:** 3 good
**Rating:** 8
**Confidence:** 3

**Summary:**

Proposes HAICO-CN, a human-AI collaboration algorithm that trains personalized noisy-label correction models to enhance collaborative decision-making. Specifically, it first clusters user noise label patterns, and then onboards a user to assign them to a cluster. Then the cluster-specific model is used to combine and refine the human and model predicted label. Results are presented using several multi-rater datasets, demonstrating improvements over baselines.

**Strengths:**

– The problem setting is interesting and of real-world importance

– The proposed method is simple, intuitive, and appears highly effective while being reasonably efficient

– The paper does a good job of reviewing and comparing to prior work

– The set of metrics defined are useful, and the experimental results are comprehensive

**Weaknesses:**

– While the paper considers its experiments on CIFAR-10N, Fashion-MNIST-H, and Chaoyang as “real-world”, I’m not entirely convinced that is appropriate since even for these, it simulates a test set for each new user by estimating a noise transition matrix. I agree that the CIFAR10N to CIFAR10H is more  “close to the real world” as the paper acknowledges – I would recommend an expanded discussion of the actual realism of the experimental setup, including the underlying assumptions, and cases wherein these may not hold.

– Unless I missed it, the paper lacks a few important ablations eg. what is performance without performing noisy label augmentation?

– The paper does not provide a principled way to select the number of clusters K. It claims that fuzzy K-Means is robust to this, but this claim is not validated. In the paper’s real-world experiments, K=3 simply works well because of a reduction in the number of users/cluster to train a cluster-specific model, which is simply an artifact of the experimental design. It would be helpful to see i) what K values work well with more data ii) what a principled way to select K might be, and if Fuzzy KMeans is indeed robust to this choice.

– I found the presentation of the approach rather complex and difficult to follow. For instance, Eq. 2 presents a complex per-user feature vector construction strategy without any particular justification/intuition. Similarly, it would be helpful to summarize the intuition behind the crowdlab consensus labeling strategy employed in Eq. 1.

– The paper would be strengthened by a deeper analysis of the noisy label patterns learned for the real-world data – while the results presented in Fig 5 are helpful, it would be nice to qualitative visualize examples of some of the labeling biases identified by the user clustering and corrected by the algorithm.

– It would be interesting to also discuss the applicability of the proposed method to settings beyond multiclass classification eg. would a similar method be applicable in a multilabel setting, where each image has multiple possible latent labels?

**Questions:**

Please address the weaknesses listed above – I would be happy to raise my rating appropriately.

---

> ### Author Response · Authors · 2023-11-23
> **Reply to reviewer SoV5**
>
> We sincerely appreciate the time and effort you dedicated to reviewing our paper. Below, you will find the responses and explanations for the questions raised during the review process.
>
> **While the paper considers its experiments on CIFAR-10N, Fashion-MNIST-H, and Chaoyang as “real-world”, I’m not entirely convinced that is appropriate since even for these, it simulates a test set for each new user by estimating a noise transition matrix. I agree that the CIFAR10N to CIFAR10H is more “close to the real world” as the paper acknowledges – I would recommend an expanded discussion of the actual realism of the experimental setup, including the underlying assumptions, and cases wherein these may not hold.**
>
> >Thank you for the insightful feedback. We agree and will make two changes:
>
> >1) We will add the following paragraph at the end of the Section 4.2 Experiment Setup:
>
> >It is important to acknowledge that while our method aims to retain the patterns of noise from annotators, the test sets for Fashion-MNIST-H and Chaoyang remain a simulation that may differ from actual annotator inputs. Our experimental setup with CIFAR10N and CIFAR10H datasets more accurately reflects real-world conditions, as these datasets incorporate crowdsourced labels in both training and testing phases.
>
> >2) We will update the term of “real-world data” to “public data” and update the title of Section 4.5 from “Real-World Results” to “Public Dataset Results”
>
> ---
>
> **Unless I missed it, the paper lacks a few important ablations eg. what is performance without performing noisy label augmentation?**
>
> >Noisy label augmentation is essential to the HAICO-CN model. After clustering, each cluster contains only a subset of training data, i.e. samples from a subset of annotators assigned to the cluster, which is insufficient for training cluster-wise models because of its potential small size or the absence of labels for certain data samples. The proposed noisy-label augmentation is designed to address this gap. Please see Section 3.2.2 for more details.
>
> >Closely related to this inquiry, we explored the impact of the number of times the augmentation was applied in Table 9. The results indicate that even a single application of noisy-label augmentation yields high accuracy on the CIFAR-10N dataset. Table 9 is replicated here for reference.
>
> | n\_aug | Accuracy |     | A+       | A-       |     | RA+      | RA-      |
> | :----: | :------: | :-: | :------: | :------: | :-: | :------: | :------: |
> | 1      | 0\.98897 |     | 0\.95300 | 0\.00402 |     | 0\.99580 | 0\.00420 |
> | 3      | 0\.98915 |     | 0\.95421 | 0\.00403 |     | 0\.99579 | 0\.00421 |
> | 5      | 0\.98922 |     | 0\.95221 | 0\.00354 |     | 0\.99630 | 0\.00370 |
>
> ---
>
> **The paper does not provide a principled way to select the number of clusters K. It claims that fuzzy K-Means is robust to this, but this claim is not validated. In the paper’s real-world experiments, K=3 simply works well because of a reduction in the number of users/cluster to train a cluster-specific model, which is simply an artifact of the experimental design. It would be helpful to see i) what K values work well with more data ii) what a principled way to select K might be, and if Fuzzy KMeans is indeed robust to this choice.**
>
> >We appreciate the reviewer's insightful comments. A principled way to select K is based on one of the following ideas::
>
> >Utilize a quantitative assessment of cluster quality, selecting the K that produces the highest score using the training users. The following table presents the silhouette scores for K values ranging from 2 to 10 in the simulation experiment with CIFAR10. Notably, the case with K=3 exhibits the highest score, aligning with the visual quality measure for the specified rationale.
> | K   | Silhouette Score |
> | :-: | :--------------: |
> | 2   | 0\.34752         |
> | 3   | 0\.55195         |
> | 4   | 0\.37054         |
> | 5   | 0\.18677         |
> | 6   | 0\.00565         |
> | 7   | 0\.00637         |
> | 8   | 0\.00194         |
> | 9   | 0\.00469         |
> | 10  | 0\.00281         |
>
> >Use the elbow method to select K, as shown in the elbow plot for the experiment with CIFAR10N here: https://ibb.co/0MfpwsW
>
> >We can also visually inspect the cluster quality with the cluster-wise transition matrix, as shown in the example for the Fashion-MNIST-H experiment: https://ibb.co/6B01YXr

---

> ### Author Response · Authors · 2023-11-23
> **Reply to reviewer SoV5 (part 2)**
>
> **I found the presentation of the approach rather complex and difficult to follow. For instance, Eq. 2 presents a complex per-user feature vector construction strategy without any particular justification/intuition. Similarly, it would be helpful to summarize the intuition behind the crowdlab consensus labeling strategy employed in Eq. 1.**
>
> >Regarding the intuition of crowdlab, we will add the following sentence following Eq.1:
>
> >Crowdlab works in two steps. In the first step, it estimates a consensus by majority vote using training images. In the second step, it trains a classifier using the initial consensus and obtains predicted class probabilities for each training example. After that, Crowdlab uses these predicted probabilities along with the original annotations from raters to improve the consensus, creating an ensemble as described in Equation 1 in the paper.
>
> >Regarding the intuition of Eq. 2, we will add the following sentence:
>
> >As annotators label a subset of the images from the training set, the provided labels belong to different images. Yet, to cluster all annotators in a single pass, each label set is arranged in a uniform format as presented in Equation 2 to enable the clustering algorithm to determine noisy label patterns present.
>
> ---
>
> **The paper would be strengthened by a deeper analysis of the noisy label patterns learned for the real-world data – while the results presented in Fig 5 are helpful, it would be nice to qualitative visualize examples of some of the labeling biases identified by the user clustering and corrected by the algorithm.**
>
> >We appreciate the suggestion. Please see below the suggested cluster-wise visualization, including the noisy pattern and examples. We will add these images into the supplementary material of the paper.
>
> >* K=3 in CIFAR10 simulation: https://ibb.co/q7cFqBp
> >* K=3 in Chaoyang experiment: https://ibb.co/HNRshgD
> >* K=3 in Fashion-MNIST-H experiment: https://ibb.co/6B01YXr
>
> ---
>
> **It would be interesting to also discuss the applicability of the proposed method to settings beyond multiclass classification eg. would a similar method be applicable in a multilabel setting, where each image has multiple possible latent labels?**
>
> >We appreciate the suggestion! We will add the following paragraph into the paper:
>
> >Aligned with the existing body of literature on human-AI collaboration works, HAICO-CN centers its attention on the domain of multi-class classification. We aim to expand HAICO-CN’s applications to different problems, like multi-label scenarios. A challenge we foresee is estimating noise transition matrices in the multi-label setting. To overcome this, we intend to learn from recent literature [1, 2] for effective solutions to this specific challenge.
>
> >[1] S. Li, X. Xia, H. Zhang, Y. Zhan, S. Ge, and T. Liu, ‘Estimating Noise Transition Matrix with Label Correlations for Noisy Multi-Label Learning’, in Advances in Neural Information Processing Systems, 2022, vol. 35, pp. 24184–24198.
>
> >[2] S. M. Kye, K. Choi, J. Yi, and B. Chang, ‘Learning with noisy labels by efficient transition matrix estimation to combat label miscorrection’, in European Conference on Computer Vision, 2022, pp. 717–738.

---

### Official Review · Reviewer_Ub55 · 2023-11-05

**Soundness:** 2 fair
**Presentation:** 3 good
**Contribution:** 2 fair
**Rating:** 5
**Confidence:** 4

**Summary:**

The paper introduces HAICO-CN, a human-AI collaborative method aimed at enhancing joint decision-making by personalizing models to individual user noise patterns. Utilizing a cluster-wise noisy-label augmentation technique, HAICO-CN trains models tailored to specific user groups identified within multi-rater datasets. The method's effectiveness is validated empirically across different datasets, outperforming current human-AI collaboration approaches.

**Strengths:**

1. **Targets on an interesting problem**:
The author chooses to focus on an important issue. As AI systems become increasingly integrated into everyday tasks, human-machine collaboration becomes critical and may have many applications.

2. **Focus on Personalization**: The authors' approach to personalizing AI models to individual users is a significant strength of the paper. Personalization is key to the next generation of AI tools, and the authors' work targets how this can be achieved in the context of noisy data and decision-making.

3. **Simplicity and Effectiveness of the Proposed Method**: The simplicity and intuitiveness of the HAICO-CN method are notable strengths. The authors have developed a technique that does not rely on overly complex algorithms or require extensive computational resources, which enhances its accessibility and potential for widespread adoption.

4. **Quality of Writing**: The paper is generally well-written and easy to understand. The authors have structured their arguments logically, making the paper accessible to readers with varying levels of expertise in the field.

**Weaknesses:**

1. **Insufficient Experiments for Comparative Analysis**:
   The paper could be strengthened by including a more comprehensive set of experiments that compare HAICO-CN with popular baselines known for their effectiveness in learning with noisy labels For example, DivideMix (ICLR20), ELR (NeurIPS20), CausalNL (NeurIPS21), C2D (WACV23), and UNICON (CVPR22). The absence of such comparisons may lead to questions about the thoroughness of the evaluation and the generalizability of the proposed method across different noisy label learning scenarios.

2. **Lack of Comprehensive Review on Learning with Noisy Labels**:
   Given that the paper addresses the challenge of noisy labels, it would be beneficial to include a thorough review of existing methods for learning with noisy labels. This review should cover the spectrum of strategies employed to mitigate the impact of label noise and how these strategies compare to the proposed HAICO-CN method. By situating HAICO-CN within the broader context of the field, the paper would provide readers with a clearer understanding of the novelty and significance of the proposed method. Moreover, such a review could highlight how HAICO-CN contributes to or diverges from established theories and practices in noisy label learning.

3. **Potential Limitations in Technical Contribution**:
   While the paper introduces a novel cluster-wise noisy-label augmentation technique, its technical contribution may appear limited if it does not sufficiently differentiate itself from existing work. For example, the paper could benefit from discussing the recent findings from "Identifiability of Label Noise Transition Matrix" (ICML23), which also employs a cluster-based approach to infer clean labels from noisy ones.

4. **Reliance on Accurate Clustering of User Noise Patterns**:
   The effectiveness of HAICO-CN is predicated on the precise clustering of users based on their noise patterns. However, this process may be fraught with challenges. Firstly, it is possible that the clustering assumption does not hold. Specifically, even if it is satisfied, it is unknown when the cluster can be identified, and whether the proposed method can successfully identify them.  There is also the concern of temporal dynamics—users' labeling patterns may evolve due to factors such as learning, fatigue, or changes in the task context. If HAICO-CN cannot adapt to these changes, the performance of the personalized models may degrade.

**Questions:**

- Could the authors elaborate on the decision to exclude certain established baselines for learning with noisy labels such as DivideMix (ICLR20), ELR (NeurIPS20), CausalNL (NeurIPS21), C2D (WACV23), and UNICON (CVPR22) from the comparative analysis?
   - How does the approach of HAICO-CN to handling noisy labels differ from or improve upon these existing methods?
   - Could the authors clarify the novel aspects of HAICO-CN's cluster-wise noisy-label augmentation technique in relation to similar approaches, such as the one presented in "Identifiability of Label Noise Transition Matrix" (ICML23)?
   - What are the unique contributions of HAICO-CN that distinguish it from other cluster-based methods for inferring clean labels from noisy data?
   - How do the authors ensure the accuracy of clustering users based on noise patterns in HAICO-CN?
   - Could the authors discuss the generalizability of HAICO-CN across different domains and types of data beyond the datasets evaluated in the study?
   - How does HAICO-CN account for the temporal dynamics of user behavior, such as learning or fatigue, which might alter their noise patterns?
   - Is there a component of continuous learning or a feedback mechanism in HAICO-CN that allows for the models to be updated in response to evolving user labeling patterns?

---

> ### Author Response · Authors · 2023-11-23
> **Reply to reviewer Ub55**
>
> We sincerely appreciate the time and effort you dedicated to reviewing our paper. Below, you will find the responses and explanations for the questions raised during the review process.
>
> **How does the approach of HAICO-CN to handling noisy labels differ from or improve upon existing methods?**
> >HAICO-CN addresses a distinct challenge related to, but not confined to, the noisy labeling problem. It is a method designed for human-AI collaboration aimed at improving joint decision-making. Within this context, we propose cluster-wise models to assist in situations where individuals may introduce varying patterns of noisy labels. HAICO-CN represents a novel methodology to tackle the complexities of human-AI interactions, building upon the foundational concept of multi-rater noisy labels.
>
> >Recent noisy labeling methods, like UNICON and C2D, have demonstrated impressive results across various levels of noise rate. Nevertheless, these approaches would require substantial modifications to effectively manage multi-rater labels or be adapted for use in a human-AI collaborative context, where models take both human and AI inputs.
>
> >Recognizing the difference of HAICO-CN compared to other noisy labeling techniques, we conducted a comparative experiment following the setup in UNICON. Specifically, we simulated five users, each introducing a 10% asymmetric noise in three class pairs (Airplane-Bird, Truck-Automobile, and Horse-Deer). Subsequently, we trained and evaluated HAICO-CN with k=3. The same experiment was repeated for 30% and 40% noise rates.
>
> >The table below presents the results. Please note that our model outperforms other methods in part because it considers inputs from both humans and AI.
>
> | Method     | Noise Rate |          |          |
> | :--------- | :--------: | :------: | :------: |
> |            | 10%        | 30%      | 40%      |
> | CE         | 88\.8      | 81\.7    | 76\.1    |
> | LDMI       | 91\.1      | 91\.2    | 84       |
> | M-Up       | 93\.3      | 83\.3    | 77\.7    |
> | JPL        | 94\.2      | 92\.5    | 90\.7    |
> | PCIL       | 93\.1      | 92\.9    | 91\.6    |
> | Dmix       | 93\.8      | 92\.5    | 91\.7    |
> | ELR        | 95\.4      | 94\.7    | 93       |
> | MOIT       | 94\.2      | 94\.1    | 93\.2    |
> | C2D        | -          | -        | 93\.7    |
> | UNICON     | 95\.3      | 94\.8    | 94\.1    |
> | Ours (K=3) | 99\.8   | 99\.6 | 99\.3 |
>
> ---
> **Could the authors clarify the novel aspects of HAICO-CN's cluster-wise noisy-label augmentation technique in relation to similar approaches, such as the one presented in "Identifiability of Label Noise Transition Matrix" (ICML23)?**
>
> >The paper "Identifiability of Label Noise Transition Matrix" (ICML23) introduces a theoretical characterization for the identifiability of the label noise transition matrix. That paper has a proof about the number of multiple noisy labels per training sample that are needed to enable the identification of the instance-based noise transition matrix. The goal of our submission is quite different. More specifically, we propose a human-AI collaborative method to allow human-AI joint decision-making by training personalized models using novel cluster-wise noisy-label augmentation techniques. In other words, we are not interested in proving the identifiability of our model – instead, we aim to propose an algorithm to train a human-AI collaborative model that can personalise to users during testing.
> ---
>
> **What are the unique contributions of HAICO-CN that distinguish it from other cluster-based methods for inferring clean labels from noisy data?**
>
> >Similar to the response to Q1, HAICO-CN tackles the challenge of human-AI collaboration, a domain related to, but distinct from, noisy-label learning. Our objective is to learn the error patterns within user clusters and devise strategies to rectify these mistakes. Then when dealing with a new user, we first identify the user's cluster (onboarding) then utilise the cluster-wise model to rectify potential errors made by the user.
> ---
>
> **How do the authors ensure the accuracy of clustering users based on noise patterns in HAICO-CN?**
> >For the simulation dataset (CIFAR10), we have the ground truth and can evaluate the accuracy of the clustering. We simulated 15 users using three noisy label patterns and users with the same noisy label patterns were all grouped into the same cluster (Figure 4).
>
> >In the case of the dataset without ground truth for clustering, visually inspecting the noise label pattern can serve as a reference for the clustering accuracy. For examples, Figure 4 on the paper and new figures below show the cluster-wise noise patterns with some qualitative visual examples for our dataset:
> >* K=3 in CIFAR10 simulation: https://ibb.co/q7cFqBp
> >* K=3 in Chaoyang experiment: https://ibb.co/HNRshgD
> >* K=3 in Fashion-MNIST-H experiment: https://ibb.co/6B01YXr

---

> ### Author Response · Authors · 2023-11-23
> **Reply to reviewer Ub55 (part 2)**
>
> **Could the authors discuss the generalizability of HAICO-CN across different domains and types of data beyond the datasets evaluated in the study?**
>
> >Like related human-AI collaboration research, including the studies by Kerrigan et al. [1] and Wilder et al. [2], HAICO-CN is focused on the image modality, especially medical imaging as its most viable application, demonstrated by our use of the Chaoyang dataset.
>
> >However, we believe that HAICO-CN can be adapted to other modalities and areas. For instance, it has potential applications in text-based tasks like news classification (AG News Dataset [3]) or content extraction (DocRED dataset [4]). We intend to further develop our method to encompass these additional applications, but this is out of the scope of the current submission.
>
> >[1] Gavin Kerrigan, Padhraic Smyth, and Mark Steyvers. "Combining human predictions with model probabilities via confusion matrices and calibration." Advances in Neural Information Processing Systems 34 (2021): 4421-4434.
>
> >[2] Bryan Wilder, Eric Horvitz, and Ece Kamar. Learning to complement humans. In Proceedings of the Twenty-Ninth International Joint Conference on Artificial Intelligence, IJCAI’20
>
> >[3] A. Gulli. The anatomy of a news search engine. In Proceedings of 14th International World Wide Web Conference, pages 880–881, Chiba, Japan, 2005.
>
> >[4] Yao, Yuan, Deming Ye, Peng Li, Xu Han, Yankai Lin, Zhenghao Liu, Zhiyuan Liu, Lixin Huang, Jie Zhou, and Maosong Sun. "DocRED: A large-scale document-level relation extraction dataset." arXiv preprint arXiv:1906.06127 (2019).
>
> ---
>
> **How does HAICO-CN account for the temporal dynamics of user behavior, such as learning or fatigue, which might alter their noise patterns? AND Is there a component of continuous learning or a feedback mechanism in HAICO-CN that allows for the models to be updated in response to evolving user labeling patterns?**
>
> >Thank you for your insightful questions and suggestions. We agree that the evolving interaction between humans and AI has the potential to alternate human behavior over time. While this issue is beyond the scope of our paper, we have acknowledged its importance in the limitation section.
>
> >That said, HAICO-CN can be modified to address this dynamic. For example, a straightforward approach is to continuously apply the onboarding process and dynamically update the assigned cluster for the user over time. This practical strategy allows for adaptation to evolving user interactions and noisy patterns.

---

### Meta-Review · Area_Chair_gUhF · 2023-12-08

**Metareview:**

This paper trains noisy-label correction models to enhance collaborative decision-making by leveraging both human and model-predicted labels.

The author provided a rebuttal on the last day of the author-reviewer discussion, resulting in insufficient discussion during this stage.

During the internal discussion, Reviewer Ub5 and Reviewer bbw3 raised critical concerns related to the usefulness and rigor of the proposed method. I asked two related questions based on these concerns, specifically:
1. How to properly select the number of clusters in real-world datasets.
2. What estimated human effort is involved in choosing the number of clusters?

After reviewing the authors' response, both Reviewer bbw3 and Reviewer Ub55 still held the same concerns.

We believe the paper would be much stronger after sufficiently addressing these concerns raised by the reviewers.

**Justification For Why Not Higher Score:**

Overall, this is an interesting paper. However, concerns about the feasibility of applying the proposed method in real-world scenarios have been intensively discussed. Additionally, the advantage of the proposed method is not clearly explained. Specifically, as pointed out by ReviewerUb55, given that labels from humans are mostly clean, many existing methods could be tweaked to achieve promising performance. We believe that the paper could be significantly strengthened by addressing these concerns in the next round.

**Justification For Why Not Lower Score:**

NA

---

### Decision · Program_Chairs · 2024-01-16

Reject